# Temporal-spatial changes and tourism efficiency coupling in Guangdong-Hong Kong-Macao Greater Bay Area

**Kaijun Wu** [1]*, **Xilin Yang**[2], **Xingfu Han**[3]

**1** School of Maritime Law and Traffic Management, Guangzhou Maritime University, Guangzhou, China,
**2** School of Psychology, Shenzhen University, Shenzhen, China, **3** Huawei Institute of Information,
Communication and Technology of Guangdong Baiyun University, Guangzhou, China

* gzwkj@126.com

Temporal-spatial changes and tourism
efficiency coupling in Guangdong-Hong Kong-
Macao Greater Bay Area. PLoS ONE 20(3):
e0313985. https://doi.org/10.1371/journal.
pone.0313985

Economics and Law, CHINA

**Peer Review History:** PLOS recognizes the
benefits of transparency in the peer review
process; therefore, we enable the publication
of all of the content of peer review and
author responses alongside final, published
articles. The editorial history of this article is
available here: https://doi.org/10.1371/journal.
pone.0313985

## Abstract

The Guangdong-Hong Kong-Macao Greater Bay Area (GBA) is an important tourist
destination in China. However, the development of regional tourism is uneven, and there
is an obvious center-periphery structure. It is significant to study the temporal and spatial
evolution of the efficiency and scale of tourism economies and the relationship between
them from the perspectives of economics and geography. The tourism scales of 11 cities
in the GBA were calculated from 2001 to 2019. The Earth Science Data Acquisition
(ESDA) method was used to investigate temporal-spatial differentiation characteristics
and changes in tourism efficiency and scale. And it used a coupling coordination model
to explore the superiority of coupling coordination and regional synergy. According to
the findings, the GBA's tourism efficiency rose steadily in fluctuations and then declined,
whereas the tourism scale developed quickly, then slowly, followed by a steady upward
trend. The efficiency and scale of tourism both have obvious spatial differences, and both
the local spatial agglomeration effect and the spatial synergy of tourism become more
and more significant. The coupling degree and coupling coordination degree between
tourism efficiency and scale have evidently improved. Tourism efficiency and scale are
currently primarily in the middle of mutual restraint and a low degree of mutual promotion.
The importance of this paper stems primarily from its emphasis on temporal and spatial
changes, as well as the coupling coordination of tourism efficiency and scale in GBA, as
the majority of the existing literature focuses on temporal and spatial changes.

## 1. Introduction

The economy of the Bay Area is essential to the current state of economic growth. As the
world's top bay areas, the New York Bay Area, the San Francisco Bay Area, and the Tokyo Bay
Area in Japan serve as central centers that drive innovation and collect radiation for economic
growth. They often possess an open economic structure, a strong agglomeration function, an
advanced international communication network, and effective resource allocation. Walker
and Schafran (2015) said that the Bay Area, which is made up of a number of sizable cities or
areas, is currently the main engine promoting economic growth and technological innovation
worldwide. Up to 60% or more of the global economy is centered in the Bay Area [1].

**Data availability statement:** The source of the underlying data set can be found in section 3.2.2 of the manuscript and there is a detailed explanation in Data Sources. The specific links to the underlying data required for the relevant research are as follows: (1) http://stats.gd.gov.cn/gdtjnj/index.html (2) https://gddata.gd.gov.cn/opdata/index?chooseValue=collectForm&deptCode=48 (3) https://www.censtatd.gov.hk/sc/EIndexbySubject.html?pcode=B1010003&scode=420 (4) https://www.dsec.gov.mo/zh-CN/ (5) https://partnernet.hktb.com/china/sc/home/index.html The specific links to the underlying summary data and data analysis results are as follows: https://pan.baidu.com/s/1rDvTBB0xSrrtjxyK_UkEaw?pwd=c95wExtractedcode:c95w.

**Funding:** Guangdong Province Philosophy and Social Science Planning Project, Project number: GD23SQYJ03. The funder had the role in decision to publish.

**Competing interests:** The authors have declared that no competing interests exist.

The Guangdong-Hong Kong-Macao Greater Bay Area (GBA) is an important tourist destination and source of tourists in China, with unique advantages in economy, transportation, policies, and geographic location. Nine cities in Guangdong in the GBA contributed 9.1% of China's GDP in 2017. In terms of inbound tourism, Guangdong accounts for about 60% of the mainland's overnight inbound tourists, including about 80% of Hong Kong residents, about 55% of Macau residents, about 57% of Taiwanese residents, and about 37% of foreigners [2]. On the whole, the tourism resources of GBA are diverse and abundant, and the level of tourism development is relatively high. Nine cities in Guangdong in the GBA have one World Cultural Heritage Site and seven 5A scenic spots, accounting for the majority of Guangdong Province's branded tourism resources. Hong Kong has relatively few natural resources, mainly relying on cultural resources. There is one World Cultural Heritage site, the Historic Center of Macao, which includes 22 buildings and 8 squares. Gambling is a unique tourism resource in Macao [2]. However, the internal development of the region is uneven, and there is an obvious center-periphery structure [3]. With Guangzhou, Shenzhen, Hong Kong, and Macau as the centers, the tourism economy in the southeast region develops well, and the tourism development efficiency of cities relatively far from the centers is not high and the scale of tourism is small; the difference in tourism development between cities is very significant. The wealth of tourism resources and the level of tourism development in each city differ, and there are significant regional and spatial differences in the structure, economic efficiency, and scale of development of the tourism industry.

Starting from the optimization of the tourism supply system, accelerating the transformation of kinetic energy through structural adjustment and industrial upgrading, and forming a tourism economic system with structure as the framework carrier, efficiency as the internal control, and scale as the external presentation are the fundamental and effective ways to maintain the coordinated, sustainable, and high-quality development of the tourism economy in the GBA [4]. Whether the two special administrative regions and nine prefecture-level cities in Guangdong, Hong Kong, and Macao can achieve stable and balanced development in time and space and grasp the coupling and coordination relationship between tourism efficiency and tourism scale is of great importance to the establishment of a tourism integrated development coordination mechanism in the GBA. Therefore, it is of practical significance to study the temporal and spatial evolution of the efficiency and scale of regional tourism economies and the relationship between them from the perspectives of economics and geography. It is beneficial to promote regional tourism as well as the economic development of Guangdong, Hong Kong, and Macao, as well as learn how to solve the problem of improving quality and efficiency. Based on this, the author, relying on the official statistical data, used the Data Envelopment Analysis with Banker, Charnes, and Cooper model (DEA-BCC model) and the total factor productivity index model (TFP model) to decompose and measure tourism efficiency, used the entropy method to measure the scale of tourism, and tried to explore through exploratory spatial data analysis and coupling analysis methods. The temporal and spatial differentiation of tourism efficiency and the scale of tourism in the GBA, the law of regional differences, and a coupling coordination model provide relevant suggestions for the development of regional tourism in the GBA.

## 2. Literature review

Research on regional tourism can be traced back to the 1960s. Based on geographical location theory, Walter Christeller (1964) conducted a spatial analysis of the economic and resource conditions of industrial agglomeration centers and peripheral regions in developed European countries such as France and Germany. He comprehensively examined some factors of European tourism location and pointed out that tourism development in peripheral regions

will be a key focus [5]. Britton (1980) applied geospatial and economic theories, taking Fiji as a case study, to analyze the spatial organization pattern and spatial polarization pattern of the country's tourism economy from various aspects such as political and historical background, foreign capital, and urban infrastructure [6]. Weaver (1998) used the core edge theory model to analyze the relationship between the tourism industries of the two Caribbean island countries in the master-slave island regions and proposed that tourism development reflects and enlarges the core edge relationship between regions and that regional tourism development should build a broader internal core edge relationship model [7]. Scholars also measure and evaluate regional tourism differences from different dimensions of the tourism system. For example, regarding the issue of regional differentiation of tourism resources, Cha et al. [8] and Simon Milne [9] analyzed the utilization level and spatial distribution of tourism resource elements. For the study of regional tourism competitiveness, Dwyer [10], Brent [11], Enright [12], and others measured the strength of regional tourism competitiveness through the construction of competitiveness index models. Some scholars analyze the regional tourism market from the perspective of traffic, such as McKicher using distance attenuation theory [13] and Carla Massidda and Ivan Etzousing using the GMM panel data analysis method to analyze the spatial flow, spatial correlation, and regional differentiation of the regional tourism segmented market [14].

In recent years, with the continuous development of regional tourism economies, academic circles have also conducted an overall study on the regional tourism development differences from the perspective of multi-level scale units such as zones [15], provincial regions [16], city regions [17], and county levels [18], which focus on the following two aspects:

The first is regional tourism's temporal and spatial differentiation characteristics and influencing factors. Su Jianjun analyzed the spatial economic difference in the tourism development of various municipal units in Shanxi Province and obtained the characteristics of the degree of imbalance and the difference in space and time [19]. Fang Yelin and Huang Zhenfang combined principal component analysis and exploratory spatial data analysis (ESDA) methods to conclude that the tourism development of 31 provinces and cities in China continues to show an unbalanced trend [20]. Guedes and Jim é Nez analyzed the cyclical asymmetric spatial pattern of cultural tourism on the Portuguese mainland, revealed the cumulative attraction mode of regional tourism development based on hierarchical network topology, and led the tourism industry from strategic central cities to peripheral cities [21]. Sarri ó n-Gavil á Nden, etc. applied Geographic Information Science (GIS) and exploratory spatial data analysis technology to the analysis of regional tourism flow in an autonomous region of Spain and used Moran's I spatial autocorrelation global index and a Lisa clustering map to find the long-term regional imbalance of tourism development between coastal areas and inland areas [22]. Song Huilin and Ma Yunlai used Moran's I to reveal the agglomeration tendency and the spatial differences between provinces in areas with similar tourism levels in China [23]. According to Panasiuk, in the modern economy, personnel carrying information and knowledge are important production factors in the development of regional tourism economies [24]. Rom O et al. conducted a spatial econometric analysis on the spatial-temporal differences of regional tourism in Europe and revealed the close relationship between territorial characteristics and regional tourism, as well as the characteristics of the spatial correlation [25]. Some scholars, such as Ahlert [26], Dritsakis [27], and Cárdenas-Garca [28], have studied the external effects of tourism development in different regions on local national income, employment, and economic growth. Lu Lin and Yu Fenglong used tourism foreign exchange income indicators to calculate absolute and relative differences in the regional tourism economy, revealing spatial differences in the tourism industry's status and the level of the tourism economy in their study area and explaining the main factors

that caused them [29]. Yu Qiuyang and Yan Xin used the Gini coefficient, the coefficient of variation, the convergence and divergence tests, and multiple regression analysis to identify the spatiotemporal differentiation characteristics, the convergence trend, and the most critical factors influencing the former two of Jiangsu Province's regional tourism economy from 2000 to 2017 [30]. Wang Xinyue and Meng Fanqing applied the coefficient of variation and geo-detector method to explore the different characteristics and influencing factors of the spatial distribution of tourism development in 33 popular tourist cities in China [31].

The second is the dynamic evolution of the spatial structure of regional tourism. Zhang Jinhe and Zhang Jie analyzed the basic characteristics and spatial autocorrelation of China tourism flow from the perspective of the spatial field effect based on the relevant sampling data of China tourism from 1999 to 2003 [32]. Yu Fenglong and Huang Zhenfang used the inbound tourism revenue data from 1996 to 2011 and the Malmquist index and economic barycenter index to measure and analyze the tourism differences between China's coastal regions and their spatial-temporal pattern evolution at three spatial scales: belt, provincial, and municipal [33]. Guo Yongrui, etc., comprehensively used the ESDA method and found that the local spatial structure of China's inbound tourism and its degree of autocorrelation are very stable and path-dependent [34]. Wang Jun and Xujinhai effectively analyzed the spatial relevance and effect of tourism development provinces in China from 2000 to 2015 based on the social network analysis method (SNA) and the revised gravity model [35].

In addition, as for tourism efficiency and tourism scale, research focuses on the operating efficiency of tourism companies or the tourism industry. The research content focuses on the calculation of tourism efficiency, the characteristics of regional differences and their influencing factors, etc. Research methods are mostly based on data envelopment analysis (DEA). In terms of tourism efficiency, it mainly analyzes the trend characteristics, spatial differences, and influencing factors of tourism enterprise tourism efficiency or tourism industry efficiency. For example, Morey and Dittman used the DEA method to evaluate the operating performance level of 54 privately owned American hotel chains [36]. Köksal and Aksu evaluated the relative operating efficiency of 24 travel agencies operating internationally in Turkey with the DEA method [37]. Barros analyzed the efficiency of tourism hotels in Portugal and found that most hotels were inefficient but still achieved improvements in efficiency [38]. Sun Panpan and Xia Jiechang used the DEA and spatial statistical analysis methods to study the spatial structure and spatial effect of tourism industry efficiency in 31 provincial regions of China from 2000 to 2012 and found the important impact of tourism quality on it. In terms of tourism scale, the research mainly focuses on the level of tourism industry development scale and the scale effects it brings [39]. For example, Jenkins studied the impact of tourism development scale on tourism projects in developing countries [40]. Zhao Lei et al. studied the scale differences and ranking system between domestic and inbound tourism in Jiangsu Province, China, from 2000 to 2009 [41]. Gong Yan calculated the static and dynamic efficiency of tourism in the upper, middle, and lower reaches of the Yangtze River economic belt and used the Tobit model to study the influencing factors of tourism efficiency [42]. Yuan Li and Sun Gennian analyzed the scale differences of inbound tourism in 21 regions, cities, and prefectures of Sichuan Province, China, from 1997 to 2015 by using ranking scale distribution theory, the coefficient of variation, and the Gini coefficient index [43].

At present, the research on the development of regional tourism in the GBA is not comprehensive enough. Only a few scholars have studied the relationship between tourism industry development and location conditions and its spatial characteristics [44,45], tourism enterprise innovation [46], etc. There are few detailed studies on the spatial-temporal differentiation and coordinated development of tourism efficiency and scale in the GBA.

In conclusion, on the one hand, the literature and research on the variations in regional tourism development and spatial structure are quite thorough and have produced a wealth of research findings. However, most of the existing studies take one of the temporal evolution or spatial differentiation and structural characteristics of regional tourism differences and start with the dynamic time series data or static spatial structure section data to analyze the spatial economic effect and collaborative development mechanism of regional tourism development, cutting the dynamic combination of the temporal and spatial attributes of regional tourism development. Therefore, it is limited to revealing the spatial-temporal differentiation and dynamic evolution of regional tourism. On the other hand, there are some research results on tourism efficiency and tourism scale, focusing on the development trend, spatial differences, influencing factors, and external effects of the two, but there are few studies on the spatiotemporal evolution and coupling analysis of tourism efficiency and tourism scale from the time and space dimensions. Therefore, the research on the spatiotemporal differentiation, evolution, and coupling coordination analysis of tourism efficiency and scale in regional tourism development is worthy of in-depth exploration and analysis and has significant research space and value.

Based on the literature review above, draw on the research of Fang Shimin et al. on the regional tourism differences and coordination in the Yangtze River Economic Belt [4]. This article is based on ESDA [47], using Moran index and LISA clustering to conduct static and dynamic analysis on the tourism efficiency and scale of GBA from 2001 to 2019, exploring the spatiotemporal changes and coupling coordination mechanisms of GBA tourism efficiency and scale.

## 3. Methodology

### 3.1. Research method

**3.1.1. DEA-BCC model.** The DEA is one of the nonparametric methods used to analyze the efficiency (or performance) evaluation of individual units. The main idea is to determine the relatively effective production frontier with the help of related linear programming and statistical data based on the fixed input or output of multiple decision-making units (DMU) and to compare the degree of deviation of the DMU from the frontier to judge the relative effectiveness [48]. In the tourism industry, it can be used to measure the efficiency of tourism enterprises or destinations. There are four main steps to using the DEA-BCC model to measure tourism efficiency: (1) Determine Decision-making Unit (DMU): It is necessary to determine the tourism enterprise or destination to be evaluated as the DMU. These DMUs should have similar characteristics and objectives for effective comparison. (2) Select input and output indicators: Based on the characteristics of the tourism industry, choose appropriate input and output indicators. Investment indicators may include tourism capital investment, human resources, tourism resources, etc.; output indicators may include tourism revenue, the number of tourists, tourist satisfaction, etc. (3) Building the DEA-BCC model: Based on selected input and output indicators, construct the DEA-BCC model. This model will calculate the efficiency score for each DMU, with higher scores indicating higher efficiency. (4) Solve the model and analyze the results. Use appropriate software or algorithms to solve the DEA-BCC model and obtain the efficiency score for each DMU. Then, analyze the results to identify the inefficient DMU and analyze its causes. This paper uses the output-oriented DEA-BCC model to measure tourism efficiency and analyze the effective current situation of the allocation of tourism elements.

**3.1.2. DEA-Malmquist productivity index model.** The DEA-Malmquist productivity index model, based on data envelopment analysis, can not only effectively decompose the total

factor productivity of tourism but also reflect the temporal trend of tourism efficiency based on panel data, which is referred to as the DEA-MI. It reveals the main factors leading to its change, and the specific expression is as follows [48]:

$$M_0\left(x_t, y_t, x_{t+1}, y_{t+1}\right) = \sqrt{\frac{D_0^t\left(x_{t+1}, y_{t+1}\right)}{D_0^t\left(x_t, y_t\right)} \times \frac{D_0^{t+1}\left(x_{t+1}, y_{t+1}\right)}{D_0^{t+1}\left(x_t, y_t\right)}} \tag{1}$$

$$M_0\left(x_t, y_t, x_{t+1}, y_{t+1}\right) = \frac{D_0^{t+1}\left(x_{t+1}, y_{t+1}\right)}{D_0^{t+1}\left(x_t, y_t\right)} \times \sqrt{\frac{D_0^t\left(x_{t+1}, y_{t+1}\right)}{D_0^{t+1}\left(x_{t+1}, y_{t+1}\right)} \times \frac{D_0^t\left(x_t, y_t\right)}{D_0^{t+1}\left(x_t, y_t\right)}} \tag{2}$$

Where: $x_t$ and $x_{t+1}$ are the input vectors of period t and t+1, respectively; $y_t$ and $y_{t+1}$ are the output vectors of period t and t+1, respectively; $D_0^t\left(x_t, y_t\right)$ and $D_0^t\left(x_{t+1}, y_{t+1}\right)$ are the distance functions of the decision-making units in period t+1, respectively, based on the technological frontier of period t+1 and t. $D_0^{t+1}\left(x_t, y_t\right)$ and $D_0^{t+1}\left(x_{t+1}, y_{t+1}\right)$ are respectively based on the technological frontier of t+1 period, t period the distance function to the decision-making unit in period t+1; $M_0\left(x_t, y_t, x_{t+1}, y_{t+1}\right)$ represents the total factor productivity index (TFPCH). When the value of TFPCH is greater than 1, it represents an increase in total factor productivity. When it is equal to 1, it means that the total factor productivity remains unchanged, and when it is less than 1, it means that the total factor productivity is decreasing.

The first term on the right side of the equation (2) represents the change in technical efficiency (EFFCH) from period t to period t+1, and the second term represents the change in technological progress (TECHCH).

Among them, changes in technical efficiency can be further decomposed into changes in scale efficiency (SECH) and pure technical efficiency changes (PECH). So formula (1) can also be written as the following expression:

$$M_0\left(x_t, y_t, x_{t+1}, y_{t+1}\right) = \frac{S_0^t\left(x_t, y_t\right)}{S_0^{t+1}\left(x_{t+1}, y_{t+1}\right)} \times \frac{D_0^t\left(x_{t+1}, y_{t+1} / VRS\right)}{D_0^t\left(x_t, y_t / VRS\right)} \times \sqrt{\frac{D_0^t\left(x_{t+1}, y_{t+1}\right)}{D_0^{t+1}\left(x_{t+1}, y_{t+1}\right)} \times \frac{D_0^t\left(x_t, y_t\right)}{D_0^{t+1}\left(x_t, y_t\right)}} \tag{3}$$

In formula (3), VRS represents variable returns to scale, and CRS represents fixed returns to scale. $S_0^t\left(x_t, y_t\right)$ represents the scale function of period t when the technological frontier of period t is used as the reference plane; $S_0^{t+1}\left(x_{t+1}, y_{t+1}\right)$ represents the scale function of period t+1 when the technological frontier of period t+1 is used as the reference plane. The first term on the right side of the equation represents the change in SECH from period t to period t+1, and the second term represents the change in PECH from period t to period t+1.

**3.1.3. Natural breakpoint classification.** Natural breakpoint classification, namely Jenks's best natural breakpoint method, considers that the data itself has breakpoints and is a map classification algorithm that classifies data according to the characteristic. Its core concept is clustering. The range and quantity of elements between each collection of data should be as similar as possible. The clustering ending conditions are the largest variance between groups and the minimum variance within groups. The discontinuity of the classification is the discontinuity of the data set. Therefore, the internal similarity of each group is the largest, and the external dissimilarity between groups is the largest [49].

**3.1.4. Exploratory spatial data analysis(ESDA).** ESDA breaks the classical Gauss-Markov hypothesis, judges the spatial correlation of variable data, and explores the spatial interaction mechanism of economic variable data. Based on the ESDA analysis framework and method proposed by Anselin L [47]., it visualizes the spatial distribution of data and identifies outliers in spatial data, focusing on detecting the spatial agglomeration of economic

phenomena and displaying the spatial structure of the data. Its core includes the construction of a spatial weight matrix, global spatial auto-correlation, the measurement of local spatial auto-correlation, and the identification of spatial associations. Based on the construction of a spatial weight matrix and the analysis of the local Moran's I and LISA cluster maps in local spatial auto-correlation, this paper analyzes the characteristics and spatial structure changes of tourism efficiency and tourism scale in the GBA in 2001 and 2019.

1) The spatial weight matrix: The spatial weight matrix is used to select different spatial weight functions to express different understandings of different spatial relationships in the data. In general, this spatial relationship is a description of the spatial neighborhood or neighbor relationship. Generally, a binary symmetric spatial weight matrix W containing several elements $\{W_{ij}\}$ is first defined to express the proximity relationship between the spatial regions of locations. Nowadays, the construction of the spatial weight index is usually completed based on two types of features: distance and connectivity. Based on the spatial adjacency index, that is, the spatial weight index of the connectivity feature, this paper uses ArcGIS 10.6 software to construct the spatial weight matrix. According to the adjacent criterion, when the area i and the area j are adjacent, $W_{ij} = 1$; otherwise, $W_{ij} = 0$.

2) Local Moran Index Analysis: Local spatial autocorrelation analysis expresses the degree of similarity between a spatial region and its neighbors, shows local spatial association and spatial heterogeneity, and illustrates how spatial dependence changes depending on different spatial locations. The commonly used indicator is the Local Moran Index (Local Moran's I). The calculation formula is as follows:

$$I_i = Z_i \sum_{j \neq 1}^{n} \omega_{ij} Z_j \qquad (4)$$

In the formula, $I_i$ is the local Moran's I of the region, $\omega_{ij}$ is the spatial weight value, and $Z_i$ and $Z_j$ are the deviations from the average of economic variables between regions i and j, respectively. Its spatial correlation mode can be subdivided into four types. Positive spatial correlation includes high-high correlation and low-low correlation, and negative spatial correlation includes low-high correlation and high-low correlation. High-low correlation, for example, means that a spatial unit with a higher attribute value than the average value is surrounded by a neighborhood with a lower attribute value than the average value, indicating that there is obvious spatial differentiation between adjacent areas, and the other three cases can be deduced by analogy. The above content is presented in a visual way, and the Moran scatter plot diagram is obtained (Fig 1).

3) LISA cluster analysis: Since the Moran scatter plot does not test the local Moran index of each regional unit, it is necessary to use a one-sample Z test to test the statistical significance of Local Moran's I under a given significance level. Visualize the cities that pass the significance test in the form of a map, which is the LISA cluster analysis.

**3.1.5. Coupling and coordination model.** Coupling degree is an index to judge the degree of interaction between two or more elemental systems based on quantitative data [50]. According to related research, this study uses the coupling degree model to judge the degree of interaction between tourism efficiency and tourism scale. The specific calculation formula is as follows:

$$C = \left\{ f(x)g(y) / \left[ (f(x) + g(y)) / 2 \right]^2 \right\}^k \qquad (5)$$

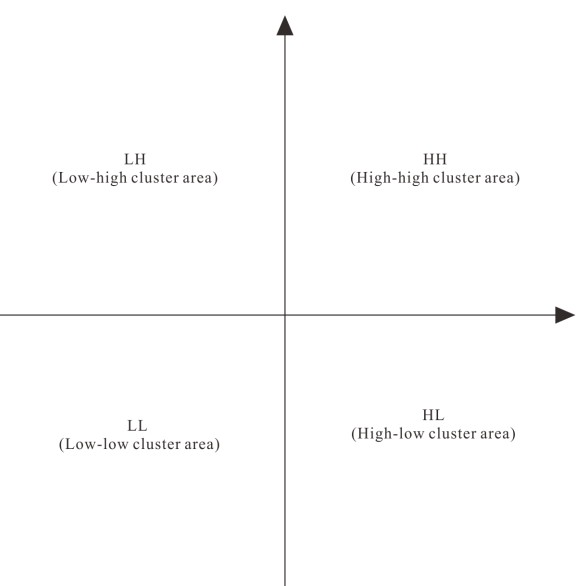

**Fig 1. The Moran scatter plot diagram.**

In the formula: C is the coupling degree between tourism efficiency and tourism scale, $0 < C < 1$, the greater the coupling degree, the better the coupling; $f(x)$ and $g(y)$ are tourism efficiency index and tourism scale index, respectively; k is the adjustment coefficient, usually $2 \le k \le 5$. Because the coupling degree model only contains two elemental systems: efficiency and scale, the value of k in this study is 2.

Coupling Coordination Degree is an index to quantitatively judge the goodness of coupling between two or more element systems, and it is a further analysis of the coupling state of tourism efficiency and scale. According to the existing research results, the calculation formula is as follows:

$$D = \sqrt{C \times T}, T = \alpha f(x) + \beta g(y) \tag{6}$$

In the formula, D is the degree of coupling and coordination between tourism efficiency and tourism scale, and T is the comprehensive coordination index of the two subsystems. Both α and β are undetermined coefficients, and the sum of the two is always 1. The two indicators of tourism efficiency and tourism scale play an equally important role in the research of this article, so α=β=0.5.

The grading of coupling coordination degree is mainly based on its value and the interaction relationship between various components within the system. Generally speaking, the higher the coupling coordination value, the tighter the coupling between the various components within the system and the better the coordination; On the contrary, it indicates weak internal interactions and poor coordination within the system. By evaluating the level of coupling coordination, we can gain a deeper understanding of the overall and coordinated status of the system, providing important references for the optimization design and management of the system [4,51].

## 3.2. Indicator selection, data sources, and robustness testing

**3.2.1. Indicator selection.** In general, the ratio of input to output is efficiency, and the same is true for tourism efficiency measurement and index selection. In terms of tourism

investment indicators, based on the three basic production factors of labor, land, and capital in classical economics, combined with the actual situation of tourism economic activities, replacement is carried out [4]. The most suitable indicator for measuring labor input is the number of tourism employees, but due to practical statistical reasons, the data availability is very low, so it is replaced with the number of employees in the tertiary industry [33]. The efficiency and scale of the tourism economy are less affected by land elements [34]. Hong Kong, Macao, and most cities have no statistics on relevant tourism land data, so they are not included in the variable index system of the study. Capital factors are the economic support for tourism development, but currently there are no economic statistical indicators directly related to tourism investment in the official statistics of GBA cities [51]. Since fixed asset investment and the provision of tourism services in the tourism economy mainly depend on star-rated hotels, A-level scenic spots, and travel agencies, and the rating system of scenic spots in Hong Kong and Macao is different from that on the mainland, there is a lot of missing data on the number of travel agencies, so according to the principles of data availability, comparability, and accurate quantification, only the number of star-rated hotels will be selected to replace the capital element as one of the investment indicators of tourism efficiency. Furthermore, there is a certain correlation between the input and performance of star-rated hotels and the demand of the tourism market [34]. The supply of tourism will change in response to changes in the tourism market's demand [51]. Whether the relationship between supply and demand is balanced will affect the efficiency of the use of input elements [4]. Therefore, selecting the number of star hotels as the input index of tourism efficiency has a corresponding reality and rationality. In terms of tourism output indicators, the most intuitive output level of tourism activities in the tourism industry is reflected in the number of tourists and tourism income, so the main output indicators use total tourism income and total number of tourists [51]. In short, the input indicators of tourism efficiency in this article are the number of employees in the tertiary industry and the number of star-rated hotels, and the output indicators are the total number of tourists and total tourism revenue.

The most intuitive manifestation of the scale of tourism is the scale of visits and the total amount of tourism consumption [4,34]. Therefore, the measurement of the scale of tourism corresponds to the tourism efficiency output index. The two data indicators of total number of tourists and total tourism income are selected to construct the measurement of the tourism scale model. Firstly, calculate the dimensionless value $S_i$ of the two indicators i, and then use the entropy method to calculate the weight $P_i$ of the total number of tourists and the total tourism income. According to the existing research results, the specific measurement formula is as follows:

$$SP_n = \sum_{i=1}^{n} P_i S_i \qquad (7)$$

**3.2.2. Data sources.** The main raw data for this study comes from the "Statistical Yearbook of Guangdong Province," the statistical yearbooks of the nine prefecture-level cities in the GBA from 2001 to 2019, the statistical bulletins of national economic and social development, and the star ratings of the official websites of the Municipal Bureau of Culture, Radio, Film, and Tourism Hotel directory, the 2001-2019 "Guangdong Tourism Yearbook," the 2001-2019 "Hong Kong Statistical Yearbook," the statistical data database of the official website of Macao Statistics and Census Bureau, and Hong Kong Tourism Net.

**3.2.3. Robustness testing.** This study empirically analyzed panel data from 11 cities in the GBA from 2001 to 2019. Ensuring data robustness is a prerequisite for avoiding false analysis. Therefore, the first step is to determine the stationarity of the data. The ADF (Augmented Dickey Fuller) method is a commonly used method to test the stationarity of data, which

**Table 1. Robustness testing of GBA tourism efficiency, scale, and its decomposition indicators.**

| Verify model settings | Tourism investment efficiency | Tourism output efficiency | Tourism scale |
|---|---|---|---|
| Standard panel ADF testing | 5.220*** | 5.058*** | 4.891*** |
| Panel ADF testing with drift | 3.031*** | 7.921*** | 4.003*** |
| Panel ADF testing with trend | 2.995** | 3.081** | 3.453*** |

** and * * *, respectively, indicate significant at the 5% and 1% levels.

directly tests whether the data has a unit root. The unit root indicates that a sequence has non-stationary properties, meaning that it has a trend or drift over time. The basic idea of the ADF method is to compare models with and without unit roots by introducing a lag term. If the value of the ADF statistic is less than some critical value, the null hypothesis can be rejected, and the sequence can be considered stationary.

The test results are shown in Table 1. Using different types of panel data, ADF tests were conducted, and the test results of tourism efficiency, tourism scale, and their decomposition indicators were reported in Table 1. The null hypothesis of the panel ADF test is the existence of unit roots, which means that the data is non-stationary. All tests reject the null hypothesis at least at the 5% level, so it can be considered that tourism efficiency, scale, and their decomposition indicators are stationary.

## 4. Results

### 4.1. The overall characteristics of tourism efficiency and scale in the GBA

#### 4.1.1. The static characteristics of tourism efficiency and scale in the GBA.

1) Static characteristics of tourism efficiency: The author used DEAP2.1 to analyze the data of the two input indicators of the number of employees in the tertiary industry and the number of star-rated hotels, and the two output indicators of the total number of tourists and the total tourism revenue. According to the DEA-BCC model, the tourism efficiency of eleven cities in GBA from 2001 to 2019 was measured, and the overall mean values of comprehensive efficiency, pure technical efficiency, and scale efficiency of the above 11 DMUs from 2001 to 2019 were calculated, respectively. After that, using the natural breakpoint classification of ArcGIS 10.6 software, the overall average values of comprehensive efficiency, pure technical efficiency, and scale efficiency were divided into five efficiency levels: superiorly high level, slightly high level, medium, slightly low level, and seriously low level.

The average comprehensive efficiency of GBA (Fig 2a) presents a distribution characteristic of being high in the middle and low on the outside as a whole. The critical values of the comprehensive efficiency, from low to high, are: 0.333, 0.459, 0.645, and 0.841. The average comprehensive efficiency of GBA is 0.72, which is as high as 72% of the ideal state, which belongs to the middle and high efficiency levels. Firstly, in the central part of GBA, with the exception of Dongguan, Shenzhen, and Zhongshan, all other cities are at high efficiency levels. Guangzhou, Hong Kong, and Macao have relatively developed economies, which have a certain impact on the surrounding areas, but Dongguan and Zhongshan are still at a slightly low level of comprehensive efficiency due to the institutional dividends and their own weak tourism foundation. Although Shenzhen has huge advantages in economy, system, and technology at the same time, there is too much competition for limited resources among the

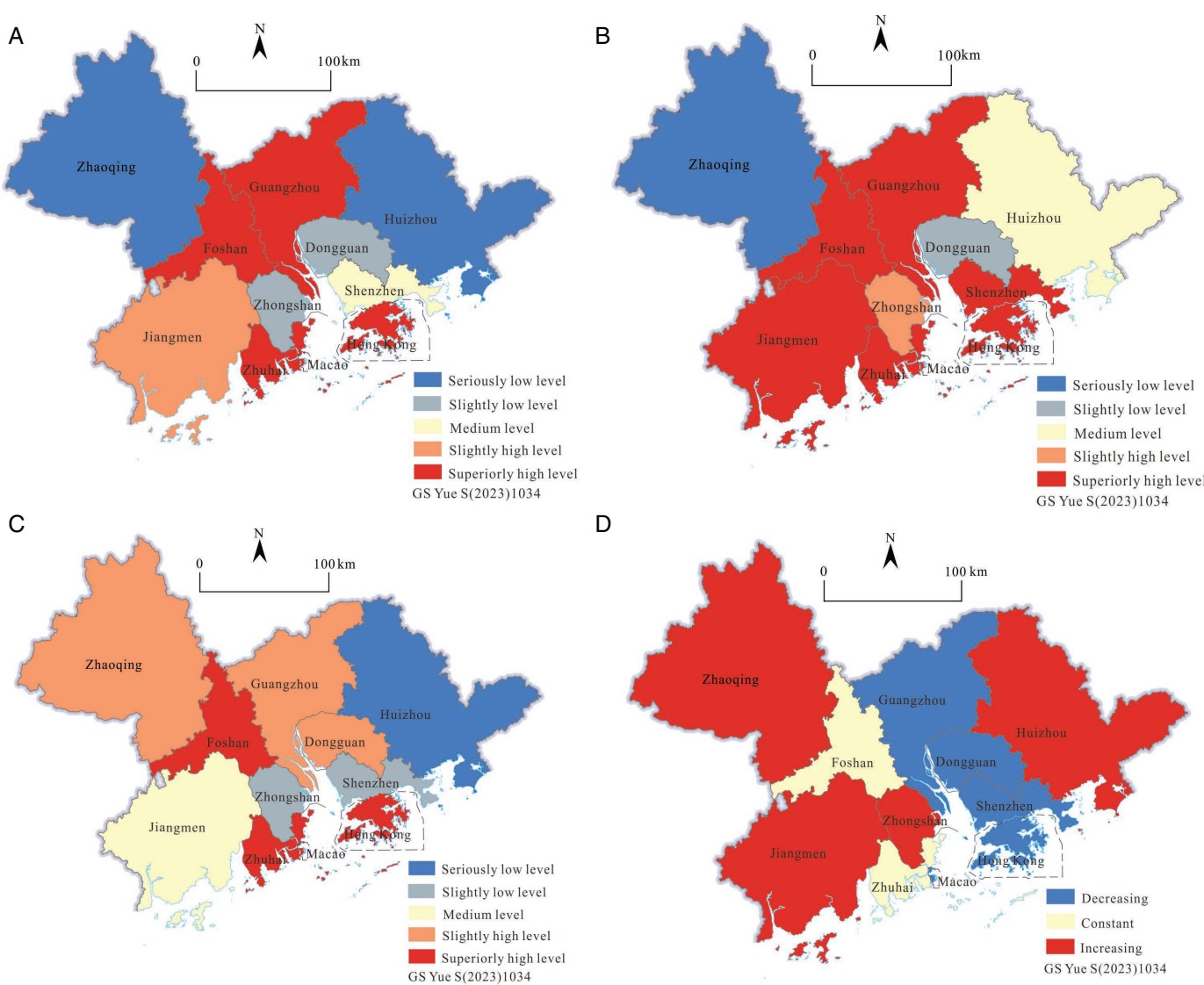

**Fig 2. Average performance of static tourism efficiency and tourism scale in the GBA from 2001 to 2019 (Drawing review number) GS Yue S(2023)1034).**

industries. The allocation of tourism technology elements is inefficient, and the long and short ones are offset, so the tourism efficiency is at a medium level. Secondly, due to Guangzhou's strong economic strength, dense population, abundant tourism resources, reasonable industrial structure, and high level of technological innovation, the overall efficiency of resource allocation in the tourism market is superiorly high. Because of many factors, such as history and national policy support, Hong Kong and Macau, as the world's two largest international free ports, have fast economic operations and a developed tertiary industry, which have driven the direction and output of tourism element input. The overall tourism efficiency of the two is superiorly high. The overall tourism efficiency of Foshan is superiorly high, too. Due to its long history and culture, rich tourism and cultural resources, large investment in tourism elements, and proximity to Guangzhou, it is affected to a certain extent by economic operations and tourism output. Zhuhai is the only city bordering Macau, and it has taken on the spillover effects of Macau's developed gambling and tourism industries. Zhuhai's tourism

cluster has obvious advantages and superiorly high tourism efficiency. Thirdly, the economic foundation in the northwest and east of GBA is relatively weak, the tourism input and output levels are not high, and the overall tourism efficiency is low. However, Zhaoqing has abundant tourist attractions, Huizhou's rapid economic development, and abundant tourism resources in the Gulf. Therefore, there is still great potential for tourism development to be stimulated and utilized. Fourthly, Jiangmen, located in the southwestern part of GBA, has a large number of overseas Chinese resources, a large scale of tourism, full utilization of technical resources, and comprehensive tourism efficiency at a slightly high level due to its administrative division and historical and cultural advantages. On the whole, the static difference in comprehensive tourism efficiency in different cities of GBA is formed by the intersection of economic development, development policy differences, diversity of tourism resources, history, and other factors.

The overall distribution characteristics of the average value of pure technical efficiency (Fig 2b) are the same as the overall efficiency. Generally speaking, it is high in the middle and low in the east and west, but the specific circumstances are different. Zhaoqing, which is located in the northwest of GBA, has a seriously low level of technical efficiency. Huizhou, which is located to the east, has a medium level of technical efficiency. The central part of GBA is at a level of medium to slightly high and above. The critical values of all levels of pure technical efficiency are 0.342, 0.476, 0.509, and 0.564, and the average value is 0.803, which belongs to the high efficiency level, indicating that the utilization efficiency of technical elements in the overall tourism economy of GBA is high, which is higher than the overall efficiency average (0.72). It can be seen that it has a certain enhancement effect on the overall efficiency. The distribution of the mean scale efficiency of GBA (Fig 2c) is quite different from the previous two, looking from west to east and from north to south. There are four cities with superiorly high efficiency levels: Foshan, Zhuhai, Macau, and Hong Kong. The slightly high efficiency levels are Zhaoqing, Guangzhou, and Dongguan; the medium efficiency levels are only Jiangmen; the slightly low efficiency levels are Zhongshan and Shenzhen; and the seriously low efficiency levels are only Huizhou. The critical values of scale efficiency at all levels are 0.579, 0.735, 0.841, and 0.975, and the overall average value is 0.883, which falls into the category of medium and high efficiency levels. The mean value of scale efficiency and pure technical efficiency are both greater than the mean value of comprehensive efficiency, and both enhance overall efficiency, but the mean value of scale efficiency is obviously greater than the mean value of pure technical efficiency, which has a greater positive effect on comprehensive efficiency. Judging from the overall situation of GBA, further improvement of tourism efficiency should focus on the progress of tourism technology efficiency and improve the level of tourism technology development and utilization.

The overall distribution characteristics of returns to scale in the GBA (Fig 2d) in 2019 are increasing in the east and west and almost decreasing in the central region. The cities with increasing returns to scale are Zhaoqing, Jiangmen, Zhongshan, and Huizhou; Foshan, Zhuhai, and Macao have the same returns to scale; and Guangzhou, Dongguan, Shenzhen, and Hong Kong have decreasing returns to scale. There are three cities with constant returns to scale, accounting for 27.27% of GBA, and their tourism element input and output have reached the optimal ratio. Cities with diminishing returns to scale accounted for 36.36%, and the phenomenon of low utilization efficiency of tourism resource elements and redundant allocation is obvious, which can appropriately reduce the investment of tourism elements and reduce the scale of development to a certain extent. Cities with increasing returns to scale also accounted for 36.36%, indicating that continued expansion of tourism investment and scale will benefit the growth of economic returns in these cities.

2) Static characteristics of the tourism scale: In order to better reflect the level difference and average situation of the tourism scale, the value of the tourism scale index is first processed to a logarithm to reduce the original value of the whole distance. Using the natural break-point classification of ArcGIS 10.6 software, the logarithmic mean value of the tourism scale index for each city from 2001 to 2019 is divided into five scale levels, namely, superiorly large level, slightly large level, medium, slightly small level, and seriously small level, with critical values at all levels. They are -4.160, -4.06, -4.93, and -3.63, and the overall mean is -3.892, which is in the middle of the large scale. According to Fig 2e, there are 3, 1, 3, 3, and 1 cities at each scale level from small to large. This means that 63.64% of cities are at the medium level or lower. It can be seen that there is still room for expansion of the tourism scale in most parts of the GBA. The high-value tourism areas are concentrated in cities with rich tourism resources and numerous historical and cultural attractions.

**4.1.2. The dynamic characteristics of tourism efficiency in the GBA.** The Malmquist Total Factor Production Index model is used to analyze the specific effects of changes in technical efficiency (EFFCH) and technological progress (TECHCH) in the GBA on total factor productivity and to deeply analyze the dynamic change process of tourism efficiency in the GBA (Fig 3 and Table 2).

From 2001 to 2019, the overall average value of the total factor productivity of tourism development in the GBA was 1.079 (greater than 1), as stated in Table 1. Compared with the previous development of tourism, the degree of tourism intensification has increased. From Fig 4, it can be seen that from 2004 to 2005, 2008 to 2009, 2014 to 2015, and 2018 to 2019, total factor productivity experienced four significant declines to varying degrees. Due to certain crisis events, technological progress and utilization were greatly weakened, and the positive impact of factor intensification on the quality of tourism development was also significant. From 2018 to 2019, the total factor productivity dropped significantly. With the gradual implementation of GBA construction, various industries have made significant investments in various tourism elements, resulting in a reduction in the utilization efficiency and intensification of technological elements on an industrial scale. The above indirectly indicates that the

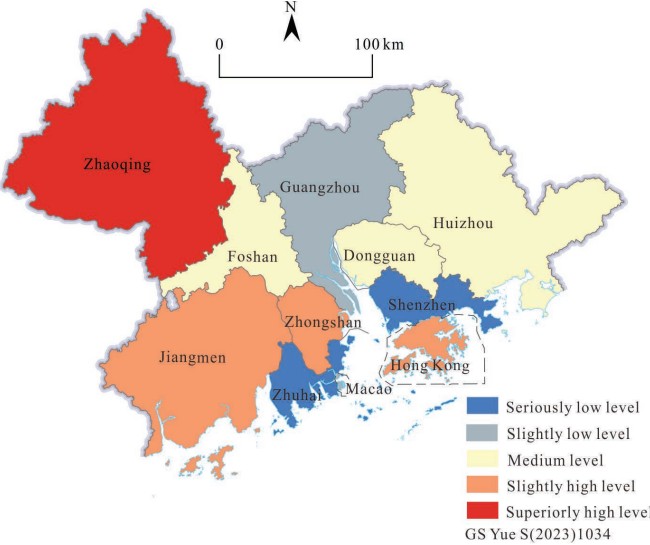

**Fig 3. The variation tendency of the Malmquist production index of tourism efficiency in the GBA from 2001 to 2019.**

**Table 2. The Malmquist production index and its decomposition of tourism efficiency in the GBA from 2001 to 2019.**

| YEAR | EFFCH | TECHCH | PECH | SECH | TFPCH |
|---|---|---|---|---|---|
| 2001—2002 | 0.949 | 1.059 | 0.893 | 1.062 | 1.005 |
| 2002—2003 | 1.025 | 0.969 | 1.003 | 1.023 | 0.993 |
| 2003—2004 | 1.079 | 1.095 | 1.153 | 0.936 | 1.181 |
| 2004—2005 | 0.873 | 1.12 | 0.846 | 1.031 | 0.977 |
| 2005—2006 | 1.027 | 1.056 | 1.038 | 0.989 | 1.084 |
| 2006—2007 | 0.991 | 1.114 | 1.03 | 0.963 | 1.105 |
| 2007—2008 | 1.12 | 0.963 | 1.004 | 1.115 | 1.078 |
| 2008—2009 | 1.007 | 1.022 | 1.046 | 0.963 | 1.029 |
| 2009—2010 | 0.965 | 1.174 | 1.01 | 0.956 | 1.133 |
| 2010—2011 | 1.079 | 1.103 | 1.09 | 0.99 | 1.189 |
| 2011—2012 | 1.033 | 1.123 | 1.02 | 1.014 | 1.161 |
| 2012—2013 | 1.001 | 1.129 | 1.02 | 0.981 | 1.130 |
| 2013—2014 | 1.001 | 1.131 | 0.997 | 1.004 | 1.133 |
| 2014—2015 | 1.079 | 0.995 | 1.031 | 1.046 | 1.074 |
| 2015—2016 | 0.888 | 1.273 | 0.998 | 0.89 | 1.131 |
| 2016—2017 | 0.966 | 1.148 | 1.003 | 0.963 | 1.109 |
| 2017—2018 | 1.003 | 1.058 | 1.034 | 0.97 | 1.061 |
| 2018—2019 | 0.897 | 1 | 0.887 | 1.012 | 0.897 |
| Average | 0.997 | 1.082 | 1.003 | 0.994 | 1.079 |

GBA is in a period of rising development, and there is a phenomenon of regional homogeneity when the scale of tourism expands, which greatly reduces the total factor productivity and affects the quality of tourism development.

The average change in comprehensive efficiency is 0.997, close to 1 (Table 2). The overall quality of tourism development tends to be stable. However, with the emphasis and in-depth construction of tourism industry development in GBA, the comprehensive efficiency rate will probably break through 1, and the effect on tourism quality will be on a positive growth trend. Except for the three periods of 2002-2003, 2007-2008, and 2014-2015, the total factor productivity of the other 15 periods is equal to or greater than the change in overall efficiency, and the average change in overall efficiency is significantly smaller than the average total factor productivity, and its contribution rate is -3%, which shows that the overall efficiency change is not a positive factor that promotes the effective development of total factor productivity.

Technological Progress Change Index: the overall average value is 1.082 (>1), indicating that the overall technological level of GBA is improving (Table 2). Its value is slightly higher than the average value of the total factor productivity index, and its contribution rate is significant, reaching 8.2%. The overall change trend of the technology change index from 2001 to 2019 is roughly the same as that of total factor productivity. It can be seen that changes in technological progress are promoting the total factor of tourism. The utilization efficiency of various elements of tourism technology is gradually increasing, which is an important factor in the positive development of productivity.

The average of the overall pure technical efficiency of GBA is 0.803, but the average of the pure technical efficiency change index is 1.003, with a contribution rate of 3%. Before the period of 2018–2019, there were only 4 years, slightly less than 1, and the overall level of pure technology utilization showed a slight fluctuation and a stable development trend. It suddenly dropped to 0.887 from 2018 to 2019, and the magnitude of the decline was similar to the decline in total factor productivity during the same period, indicating that the use of tourism

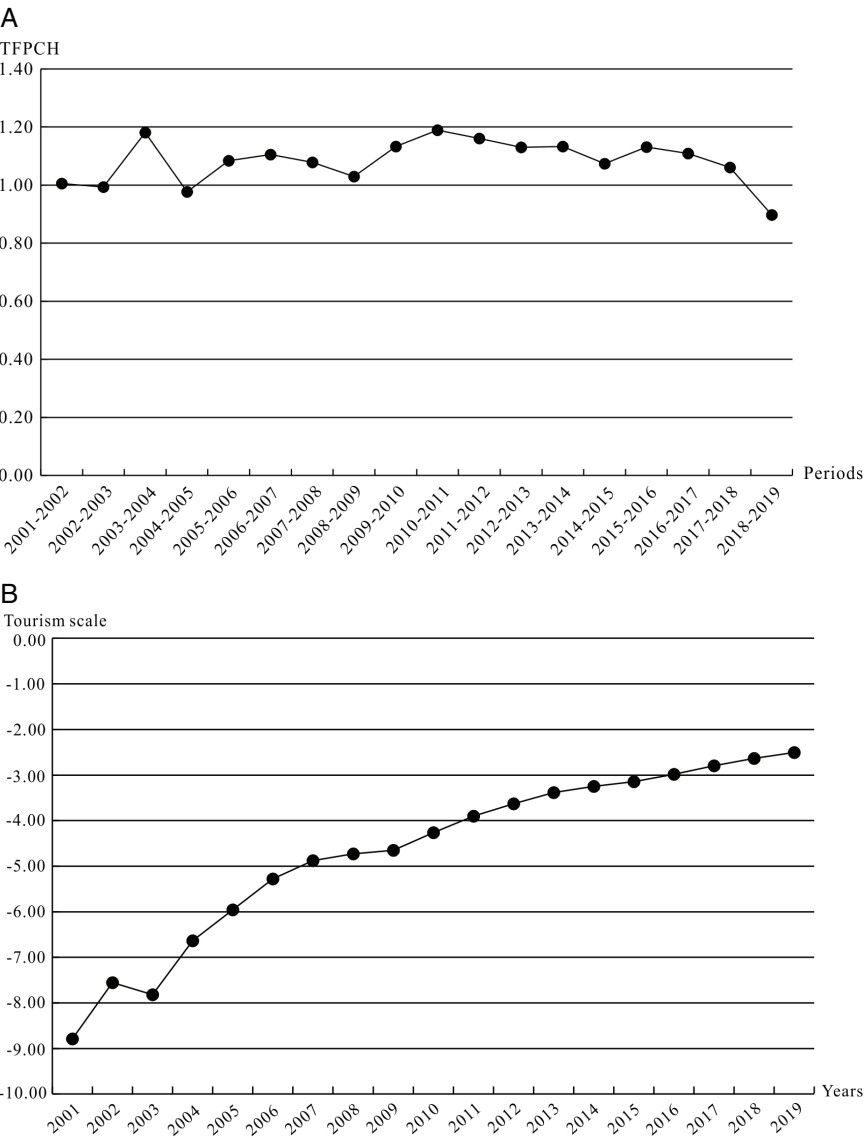

**Fig 4. Variation tendency of the tourism scale index in the GBA from 2001 to 2019.**

resources and technology in this year not only did not play a positive role but led to the efficiency of total factor utilization. And the decline in the quality of development also reflects the inconsistent foundation of tourism technology and the instability of factor levels in the GBA (Table 2).

The average scale efficiency change index from 2001 to 2019 is 0.994, which is obviously close to 1, and the average scale efficiency is 0.883 (Table 2). It can be seen that the scale efficiency level of GBA is relatively high, and the overall scale efficiency change tends to be stable. Comparing the scale efficiency change and the pure technical efficiency change index with the comprehensive efficiency change index, although the absolute value of the difference between the mean value of the three and 1 falls in the interval [0.003, 0.006], it is very close to 1, but the overall increase and decrease of the pure technical efficiency change index is greater, and the fluctuation trend of the change is more consistent with the comprehensive efficiency change

index (Table 2). Therefore, compared to the scale efficiency change index, the pure technical efficiency change has a more significant contribution to the overall efficiency change. The change in scale efficiency played a slight inhibitory effect, which resulted in a compromise effect on the overall efficiency change, and the overall efficiency of tourism development basically tended to be stable. Based on the above analysis, from 2001 to 2019, under the overall situation of small and stable fluctuations in the scale efficiency change index and significant changes in technological progress, the pure technical efficiency change index suddenly dropped by 14.22% from 2018 to 2019, and the comprehensive efficiency change index followed with a decrease of 10.57%, indicating that the utilization level of tourism technology in the GBA has declined in the past two years and the unreasonable market allocation of tourism resources is the main reason that hinders the high-quality and coordinated development of regional tourism.

At present, regional tourism in the GBA is at the intersection of the development opportunity period and the exploration period. While expanding the scale of tourism and pursuing economic efficiency, it is necessary to improve the technological level of tourism hardware facilities and introduce better service levels. High tourism talents will improve the overall efficiency of tourism in the Bay Area and give better play to the advantages of regionally coordinated development.

**4.1.3. The dynamic characteristics of tourism scale in the GBA.** Due to the significant difference in tourism scale index data among the 11 cities in the GBA from 2001 to 2009, logarithmic processing is required to calculate the overall average of the annual tourism scale index. The results indicate that, except for a slight decrease due to the impact of SARS in 2003, the tourism scale of GBA showed a steady upward trend (Fig 4).

## 4.2. Temporal and spatial dynamic characteristics of tourism efficiency and scale in the GBA

**4.2.1. Temporal and spatial dynamic characteristics of tourism efficiency.** In this study, ArcGIS 10.6 software was used to set up the spatial weight matrix and calculate Moran's I index of tourism efficiency for 11 cities in the GBA in 2001 and 2019, respectively. Based on the Z test (P < 0.05), Moran scatter maps of 2001 and 2019 were drawn, and the statistical results of the significance test with a P value of 0.05 were shown as LISA cluster to judge the statistical significance of the local correlation types and aggregation regions of each particular region.

According to the calculation result of formula (4), the tourism efficiency Moran scatter maps in 2001 (Fig 5) had a Moran's I value of 0.095 (greater than 0), indicating that the tourism efficiency of GBA in 2001 showed a positive space. Correlation: this result is more than 95% likely to be statistically significant, but its value is smaller and the degree of spatial correlation is smaller. As shown in Fig 5, most cities fall into the first and third quadrants. The number of all-type and HH-type cities is relatively large, accounting for 63.64%, indicating that most cities with a higher (or lower) than average tourism efficiency index are surrounded by cities with a higher (or lower) than average tourism efficiency index. It can be seen that the local spatial difference in tourism efficiency in the GBA is not obvious. As shown in Fig 5, Moran's I was 0.114 in 2019 (greater than 0). Tourism efficiency showed a positive spatial correlation, although spatial agglomeration was more obvious. The level of Local Moran's I index of tourism efficiency in each city in the GBA had increased, but the spatial structure had not changed much from 2001. And it can be seen that the spatial correlation of the development of tourism efficiency in the GBA has increased, but the structure is relatively stable. While the agglomeration of cities with high tourism efficiency in the GBA increases, it will drive the development of tourism efficiency in the surrounding areas through spillover effects.

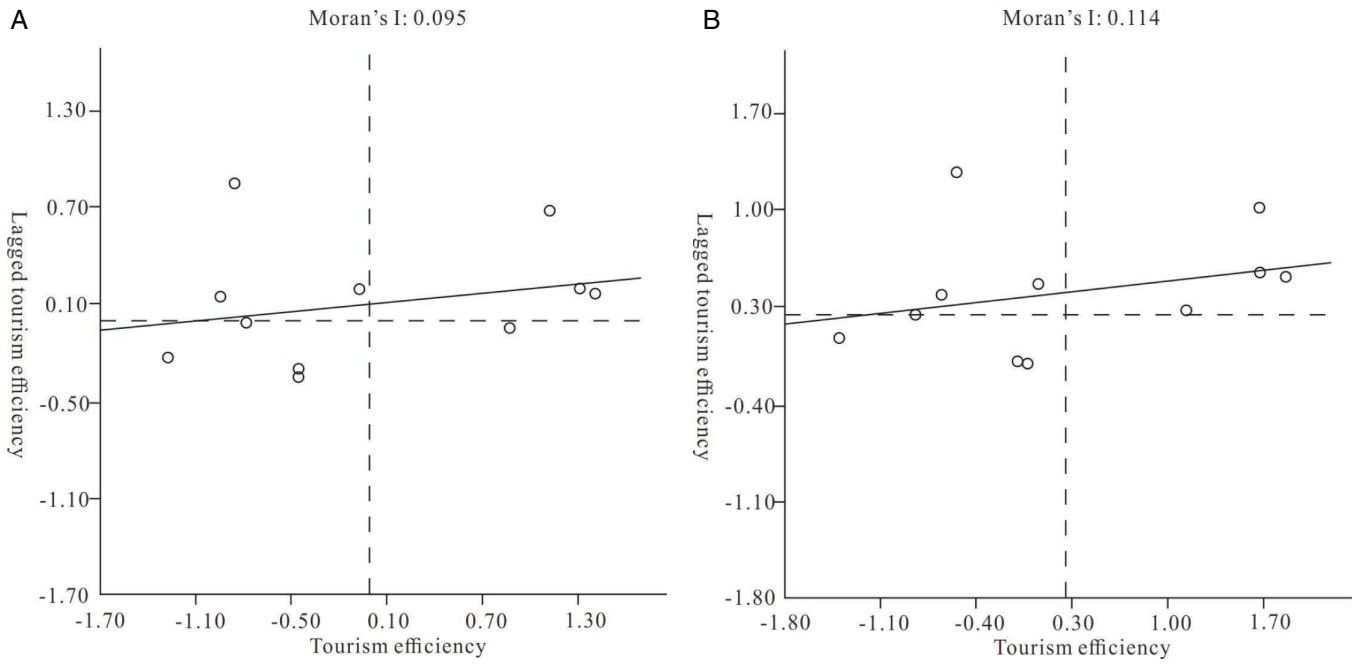

**Fig 5. The Moran scatter maps of the tourism efficiency in 2001 (left) and in 2019 (right).**

From the first-order linear fitting curve, in 2001 and 2019, some cities of the GBA all fell near the fitting curve (6), indicating that the tourism efficiency of these cities has been similar and there is a positive synergy; in 2001 and 2019, some points are far away from the fitting curve (5), which also shows that there are always individual differences in the dynamic development of regional tourism efficiency in the GBA.

The "high high" and "low low" value points in the Moran scatter maps (Fig 5) have clustering phenomena, but in the 2001 LISA cluster(Fig 6), only the H-H cluster of Foshan is more significant statistically, and the agglomeration effect of high tourism efficiency in Foshan and its surrounding areas is very prominent, which shows that the local spatial synergy of tourism development in this region is strong. Zhongshan and Hong Kong, where L-L gathers, are similarly situated, and the statistical significance of the LISA cluster in 2001 was higher. Dongguan, where H-L is clustered, and Shenzhen, where L-H is clustered, show a negative spatial correlation, indicating that Dongguan and Shenzhen have a relatively unbalanced relationship with their surrounding cities in the development of tourism. Space synergy is poor, and there are tourism resources, element utilization preemption phenomena, and crowding-out effects. The distribution characteristics of significant agglomeration effects in the LISA cluster (Fig 6) in 2019 have undergone major changes, and Jiangmen has replaced Zhongshan as a new city with significant effects. The regional agglomeration effect of H-H aggregation is more significant and appears to be a diffusion effect. The high-value point of H-H aggregation diffuses to Jiangmen on the basis of Foshan, which indicates that Foshan and Jiangmen are the high-value centers of tourism efficiency and produce spatial spillover to the surrounding areas. The collaborative role of Jiangmen and Foshan in regional tourism efficiency development was further strengthened. Shenzhen changed from L-H aggregation in 2001 to L-H aggregation in 2019. Its local spatial auto-correlation changed from negative to positive, and the local spatial difference decreased.

On the whole, the spatial relevance of the development of tourism efficiency in the GBA has been continuously strengthened, the spatial structure is relatively stable, and the local

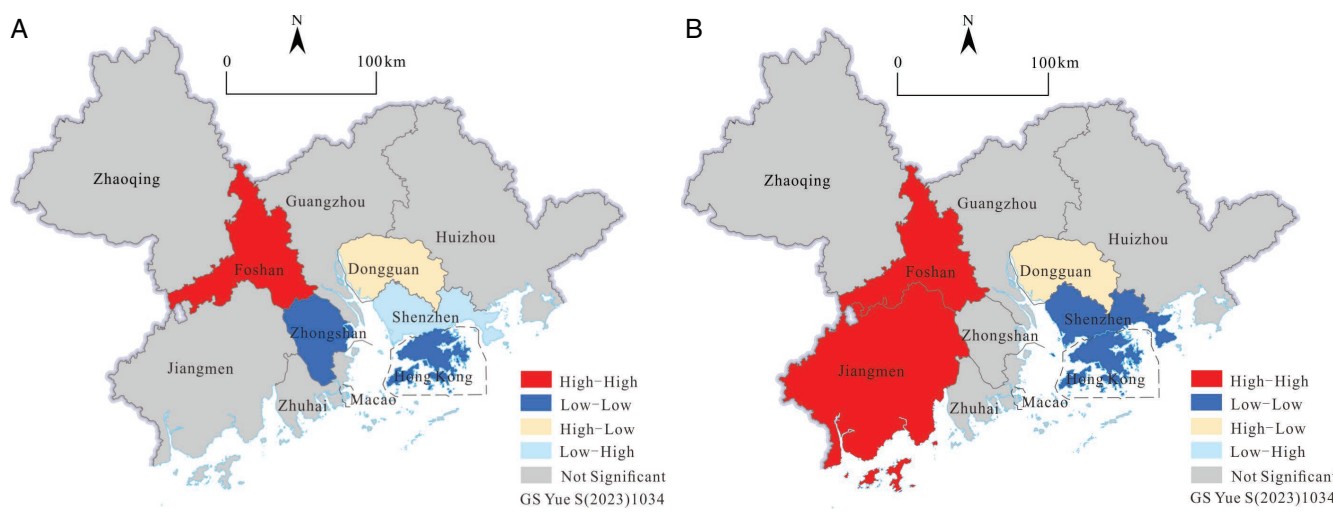

**Fig 6. The LISA cluster maps of tourism efficiency in the GBA in 2001(left) and in 2019(right) (Drawing review number) GS Yue S(2023)1034.**

spatial aggregation effect and synergy have also progressed, but there are still obvious spatial differences and local development imbalances. In the development of tourism, the GBA should strengthen the balance of factor distribution and the spatial coordination of factor utilization efficiency and avoid excessive agglomeration and homogeneity of factor inputs so as to promote the integrated development of regional tourism in the GBA.

**4.2.2. Temporal and spatial dynamic characteristics of tourism scale.** The local Moran's I index of tourism scale in 11 cities of GBA in 2001 and 2019 was calculated using ArcGIS 10.6 software, and Moran's scatter maps for 2001 and 2019 were drawn based on the Z-test (P < 0.05). Based on the significance test with a P-value of 0.05, LISA cluster were drawn.

According to the calculation result of formula (4), the Moran scatter map of the tourism scale in 2001 (Fig 7) had a Moran's I value of 0.045 (greater than 0), indicating that the tourism scale of GBA in 2001 had a spatially positive correlation, but its value was smaller and the spatial correlation degree was smaller. As shown in Fig 8, most cities are located in the first and second quadrants, with a relatively large number of L-H and H-H cities. 45.45% of cities have a low tourism scale index but high surrounding areas, while 36.36% of cities have a high tourism scale index and are surrounded by neighboring cities. It can be seen that there are significant local spatial differences in tourism scale in the GBA at this time. In 2019 (Fig 7), Moran's I is 0.245 (greater than 0); tourism efficiency still shows a positive spatial correlation, and the spatial agglomeration trend is more significant. In the Moran scatter map, 72.73% of cities fall into the first and third quadrants, which are "low-low" or "high-high" associations, indicating that the local spatial differences in the development of tourism in the GBA in 2019 are not obvious, and the scale has obvious spatial synergy. Compared with 2001, the degree of positive spatial correlation with the scale of tourism development in the GBA has increased in 2019, and the local spatial differences have become smaller.

The first-order linear fitting curves in 2001 and 2019 are obviously different in slope and intercept. It can be seen that the local spatial structure of the tourism scale of GBA has undergone significant changes over time, shifting from primarily L-H to H-H aggregation. It is mainly L-L and H-H aggregation. The fit of the first-order linear fitting curve in 2019 is slightly better than that in 2001, indicating that the difference in tourism scale between the cities of GBA has become smaller and that there is a positive synergy.

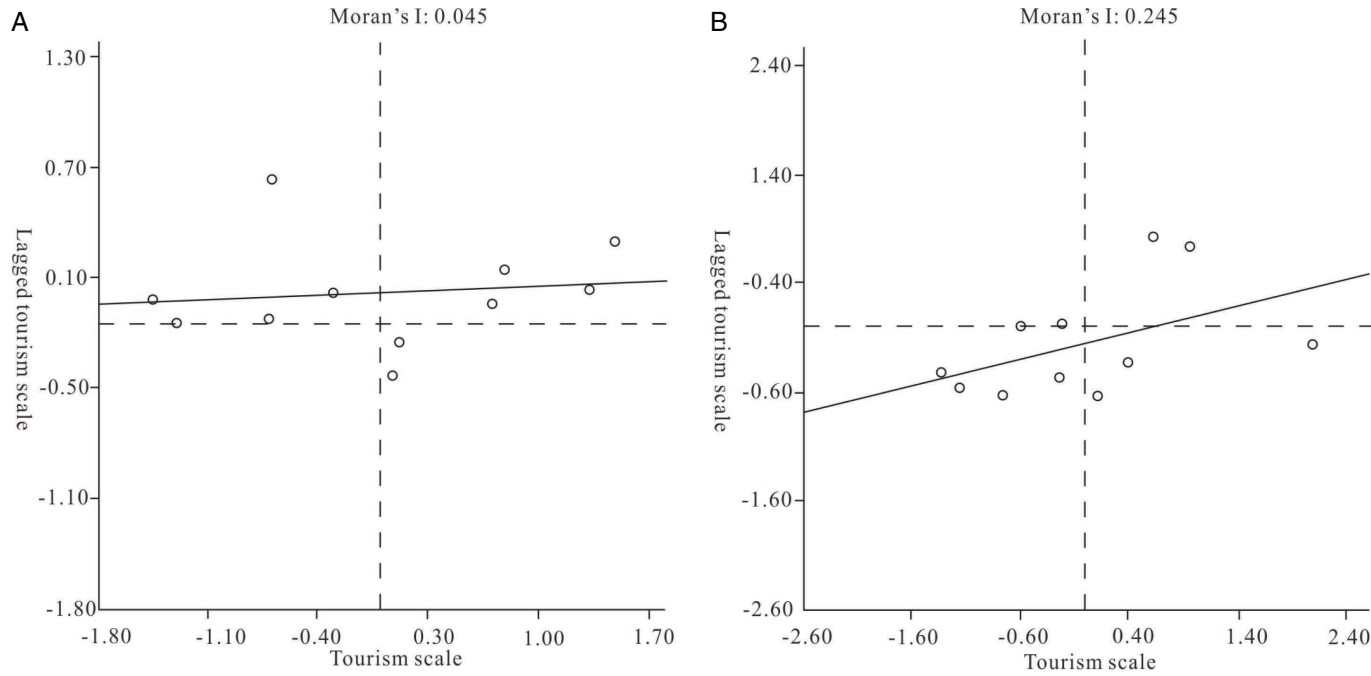

**Fig 7. The Moran scatter maps of the tourism scale in 2001 (left) and in 2019 (right).**

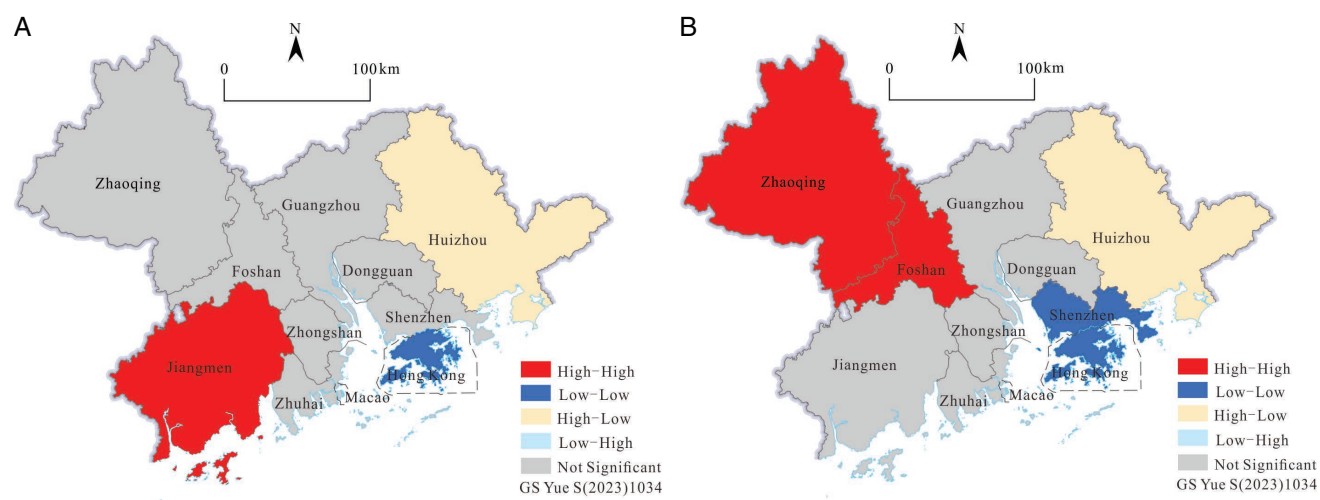

**Fig 8. The LISA cluster maps of tourism scale in the GBA in 2001 and in 2019 (Drawing review number) GS Yue S(2023)1034.**

In the 2001 LISA cluster maps (Fig 8), the statistically significant effects of local spatial agglomeration are as follows: Jiangmen, where H-H gathers; Hong Kong, where L-L gathers; and Huizhou, where H-L gathers. In the 2019 Lisa cluster maps (Fig 8), the cities with a more significant local spatial agglomeration effect are Zhaoqing and Foshan with H-H agglomeration, Shenzhen and Hong Kong with L-L agglomeration, and Huizhou with H-L agglomeration. Consistent with the analysis of the Moran scatter maps, the local spatial structure of the scale of tourism development in 2019 has changed. Specifically, the high-value and significant

area of H-H has changed from Jiangmen and its neighbors to Foshan and its surrounding areas, reflecting tourism in the GBA. With the shift of the focus toward scale development and expansion, Foshan and Zhaoqing have a significant diffusion effect on the scale development of tourism in the surrounding areas. However, at the same time, the "high-low" agglomeration effect in the eastern part of GBA is very significant. In 2001 and 2019, Huizhou's own tourism scale index maintained a relatively high level, and the surrounding area's tourism scale index was low, which was unbalanced with the surrounding area. The competitive relationship reflects the crowding-out effect on the development of tourism scale in the eastern part of GBA, with large local spatial differences and weak synergy.

On the whole, the spatial correlation of tourism-scale development in the GBA has been improved, and the local spatial agglomeration effect and positive synergy have also been enhanced. However, the local spatial structure is in the process of adjustment and change, and there are still obvious spatial differences and unbalanced local development. The GBA should balance and optimize the spatial distribution of investment in tourism elements such as technology, talent, and infrastructure and appropriately guide the spatial flow of tourism demand so as to strengthen the integration of the synergy of tourism scale development between regions.

### 4.3. The coupling and coordination relationship between tourism efficiency and tourism scale in the GBA

**4.3.1. Coupling degree spatiotemporal differentiation.** Calculate the coupling degree between tourism efficiency and scale in the GBA from 2001 to 2019 according to formula (5). The tourism efficiency index measures the total factor productivity of tourism. Combining the relevant research results with the actual situation of this study and taking 0.3, 0.5, and 0.8 as the critical values, the coupling degree (c) is divided into four types: when C = 0, there is nothing to do with tourism efficiency or scale, and the development of both is disordered. Low coupling period ($0 \leq C \leq 0.3$)—a game between the two. During the antagonistic period ($0.3C \leq 0.5$), the interaction between the two strengthens, and there is a phenomenon of mutual occupation of each other's resource space during development. During the running-in period ($0.5C \leq 0.8$), the two start to restrain and reach balance, and the coupling gradually becomes benign. Coordination coupling period ($0.8C \leq 1$): the benign coupling between the two becomes stronger and stronger. When C = 1.0, the coupling between tourism efficiency and tourism scale reaches a benign state and tends towards a new and orderly, structured development.

The author used ArcGIS 10.6 software to analyze the spatial distribution of coupling degrees in 2001, 2010, and 2019 (Fig 9). In 2001, 2010, and 2019, the average coupling degree of tourism efficiency and scale for the GBA was 0.0005, 0.028, and 0.297, respectively. The maximum coupling degree in all years was 0.279 in 2019, and the minimum was 0.0004 in 2003, and then increased year by year. On the whole, the coupling degree of GBA is in a low coupling period. Efficiency and scale in tourism are playing a game with each other, and the coupling has not entered the stage of benign coupling yet. From the perspective of the vertical time dimension, the correlation between tourism efficiency and tourism scale is gradually increasing, and the value of the coupling degree shows an overall upward trend. In 2001, the tourism efficiency of GBA had little to do with the overall tourism scale. Tourism development was in a state of disorder. Foshan had the highest degree of coupling, and its value was only 0.005. There was only a slight correlation between tourism efficiency and tourism scale. In 2010, as a whole, the GBA was still in the low coupling period, and the coupling situation had improved, but the maximum value was only 0.046, and the tourism efficiency and scale

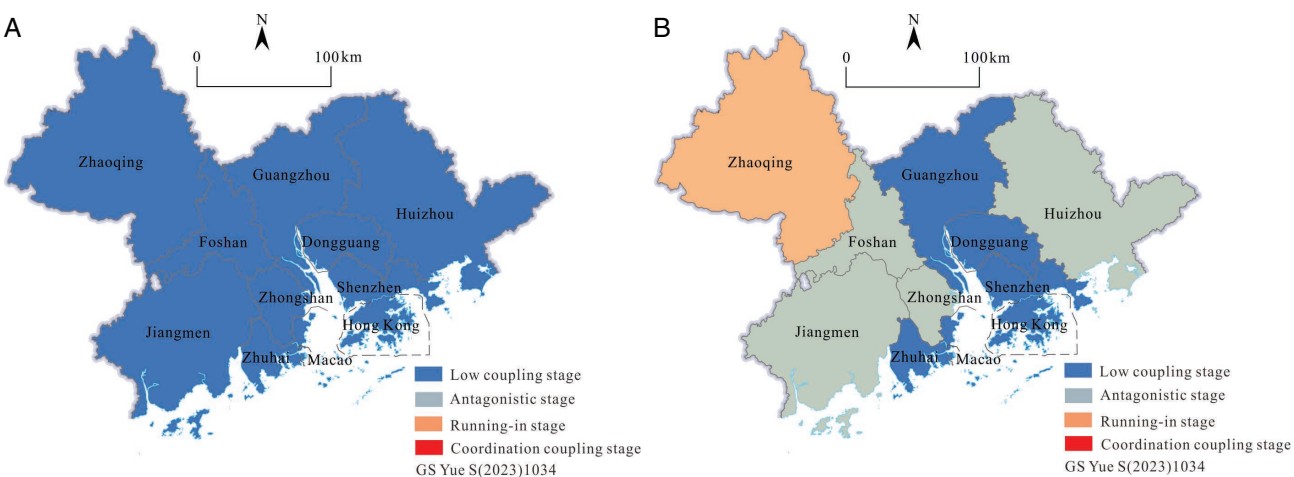

**Fig 9. Spatial distribution of the coupling degree and coordination coupling degree of tourism efficiency and tourism scale in the GBA (Drawing review number: GS Yue S(2023)1034).**

began to tend to the relevant state; in 2019, the coupling degree of tourism efficiency and scale had improved significantly, with the highest value of Zhaoqing (0.512) and the lowest value of Macao (0.151). Zhaoqing had entered a running-in period, and tourism in the GBA had begun to show a benign coupling between efficiency and scale. The overall coupling degree distribution was low in the middle and high on both sides, and regional differences were gradually appearing.

The results showed that the overall coupling between tourism efficiency and scale in GBA was increasing. However, from the perspective of specific urban cities, the specific situation was complex and changeable: although the degree of coupling between tourism efficiency and the scale of some cities was gradually increasing, they were still in a low coupling period, and the two were playing games in a disordered state. Some cities had entered an antagonistic phase of strengthening interaction, with an orderly development trend. Some cities had entered the running-in period first, but their coupling degree was still close to the antagonistic period (C = 0.5). Although they had checked and balanced each other, the trend of benign coupling was not significant. To sum up, the tourism efficiency and scale of GBA need to further strengthen the coordinated development, improve the coupling level, and promote the high-quality development of tourism.

**4.3.2. Spatiotemporal differentiation of coupling coordination degree.** Firstly, calculate the coupling coordination degree between tourism efficiency and scale in the GBA from 2001 to 2019 according to formula (6). This article divides the coupling coordination degree (D) into the following five levels: When $0 \leq D \leq 0.2$, the coupling relationship between tourism efficiency and scale is severely imbalanced. When $0.2 < D \leq 0.4$, it is a moderate disorder; when $0.4 < D \leq 0.5$, basic coordination; when $0.5 < D \leq 0.8$, moderate coordination; when $0.8 < D \leq 1$, it is highly coordinated. Secondly, use ArcGIS 10.6 software to analyze spatial distribution maps of coupling coordination degrees for 2001, 2010, and 2019 (Fig 10).

In 2001, 2010, and 2019, the mean values of the overall coupling coordination degree of GBA were as follows: 0.007, 0.128, and 0.394, showing an overall upward trend, but the overall coupling coordination level was not high and in a moderate imbalance. In 2001, the lowest value of the coupling coordination degree was Zhuhai, with a value close to 0, and the highest value was Foshan, with a value of 0.05. The development of tourism in all cities in the GBA was seriously out of balance, and the difference between cities was not significant. In 2010,

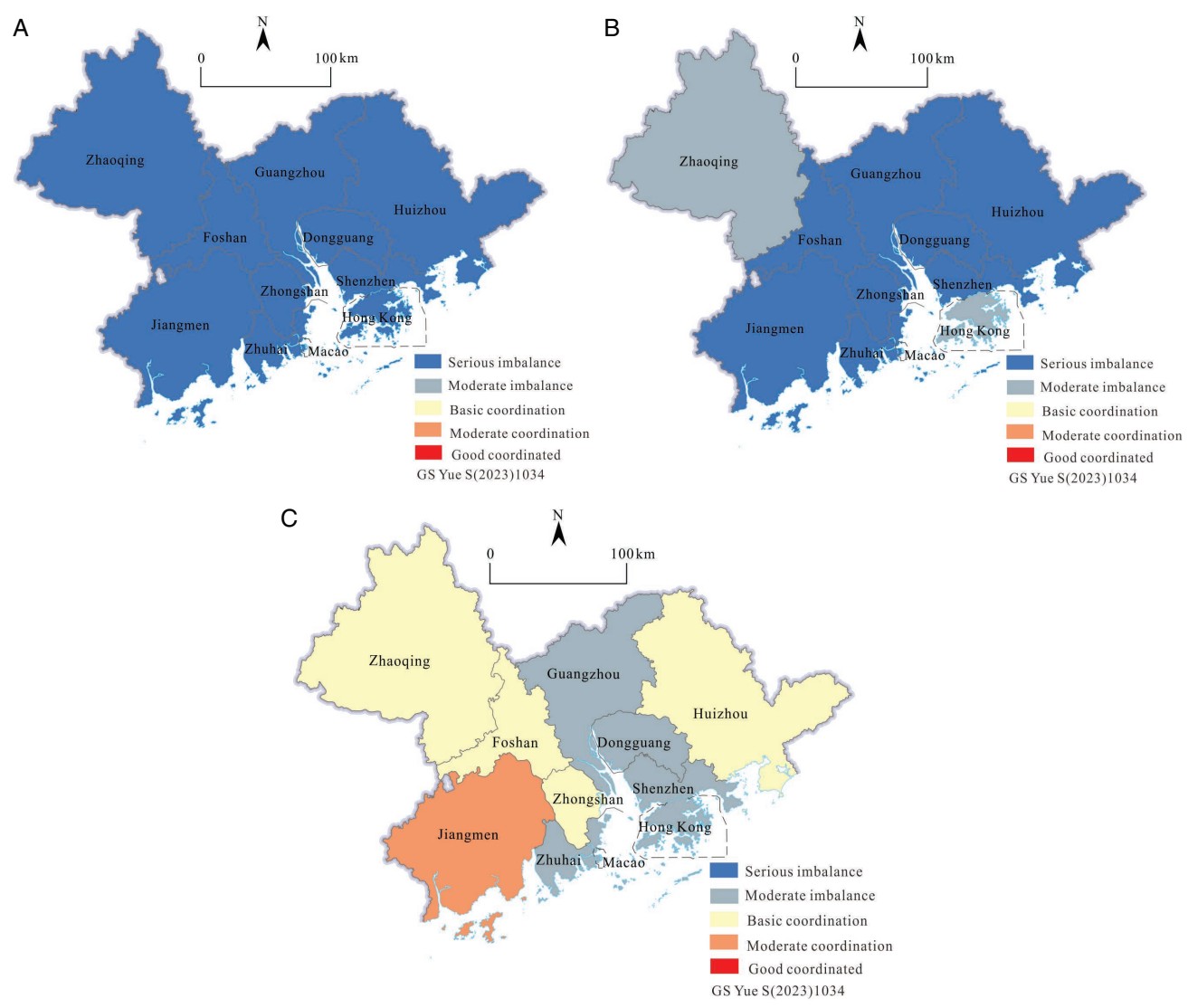

**Fig 10. The Moran scatter maps of the tourism efficiency in 2019.**

Zhaoqing and Hong Kong were the areas with the highest coupling coordination degree, which belonged to the type of moderate dissonance. 81.82% of cities were still severely misaligned. The overall level of coupling coordination was low, and spatial differences gradually appeared. The coupling coordination degree of GBA's overall tourism efficiency and scale had significantly improved in 2019. The coordination level of tourism development was medium; the low-value cities had reached a moderate imbalance; the high-value cities had reached moderate coordination; and 54.55% of the cities were in the basic coordination and moderate coordination states. The overall coupling coordination degree was low in the middle and high in the periphery, with significant regional differences. The high-value cities had a certain radiation effect on the surrounding area and had spatial differentiation.

The results show that the overall coupling degree and coupling coordination degree of tourism efficiency and scale in the GBA from 2001 to 2019 are gradually improving, but the coupling degree and coupling coordination degree of most cities are still at a medium or below level. Although the coupling distribution of tourism efficiency and tourism scale in the

GBA and the high or low value cities of the coupling coordination degree distribution do not completely overlap, from the perspective of spatial structure characteristics, the two have similar spatial differentiation; from the perspective of time evolution, the two also have a certain degree of positive correlation. In terms of the sensitivity of time and space changes compared with coupling degree, the high-value cities of coupling and coordination between tourism efficiency and tourism scale appeared earlier and spread more gradually. It can be seen that with the high quality development of tourism and the large investment in tourism economic factors in the GBA, the interaction between tourism scale and tourism efficiency will deepen rapidly, but the gradual coupling coordination reflects the limited benign coordination between them. At present, tourism efficiency and scale are in a state of medium-level mutual restriction and low-level mutual promotion, so the quality of the tourism economy and the effectiveness of regional integration are still in a state of slow development. The over investment and agglomeration of tourism core resource elements are mainly in Guangzhou, Shenzhen, Hong Kong, and Macao, which inhibits the healthy development of district tourism. As a result, the central cities themselves have not formed an effective and advanced utilization level and productivity of tourism elements and have not formed an efficient cooperation and coordinated development model with the surrounding areas. Therefore, it is necessary to adjust the allocation of resources according to local conditions and disperse the tourism resources, technology, talents, and other elements invested in the central city to other cities so that the tourism development of all cities in the region can achieve its own advantages on a better basis and promote the synergy between regions and the high-quality development of overall tourism.

## 5. Conclusions and discussion

### 5.1 Conclusions

Firstly, from 2001 to 2019, the average value of tourism comprehensive efficiency in the GBA has a static spatial difference, showing the distribution characteristics of high in the middle and low in the outside, which is formed by the interaction of economic development, development policy differences, tourism resource diversity, history, and other factors. This is consistent with the research results of Wang Zhaofeng et al. on tourism efficiency and scale in the Yellow River Basin [51]. Both scale efficiency and pure technical efficiency enhance comprehensive efficiency, but according to the mean distribution data, scale efficiency plays a greater role in supporting comprehensive efficiency. The high-value cities on the tourism scale are concentrated in cities and regions with rich tourism resources and numerous historical and cultural attractions.

Secondly, from 2001 to 2019, the overall tourism scale of GBA is on the rise, except for a slight decline in 2003; the rest of the annual increase is year by year; the development speed is fast first and then slow, and the development mode of the tourism scale has turned stable. From 2001 to 2019, the total factor productivity of tourism development in the GBA was in a dynamic fluctuation, and now it is on a downward trend. Overall, the change in comprehensive efficiency tends to be stable, which is not a positive factor in promoting the effective development of TFP. Under the overall situation of a small, stable fluctuation of scale efficiency change index and a significant change in technological progress, the level of tourism technology utilization in the GBA in the past two years has declined, and the market allocation of tourism resources is unreasonable. This is the main reason that hinders the high-quality and coordinated development of regional tourism. This is consistent with the research results of Fang Shimin et al. on the tourism scale of the Yangtze River Economic Belt [4].

Thirdly, the local spatial structure of tourism efficiency is relatively stable, while the local spatial structure of tourism scale is in the process of adjustment and change. Although the

local spatial aggregation effect and synergy between the two have made some progress, there are still obvious spatial differences and local imbalances. The GBA should strengthen the balance of the distribution of tourism elements and the spatial cooperation of the utilization of those elements so as to improve the level of tourism efficiency. We should balance the spatial distribution of investment scales in technology, talents, infrastructure, and other factors, avoid excessive concentration of factor investment in central cities, optimize the tourism scale efficiency of each city, and strive for the healthy development of regional tourism in the GBA.

From 2001 to 2019, the coupling degree and coupling coordination degree between tourism efficiency and scale in the GBA have gradually increased, but at present, the coupling degree and coupling coordination degree of most cities are still at the medium level or below. The overall coupling coordination degree was low in the middle and high in the periphery, with significant regional differences. Although the coupling distribution of tourism efficiency and scale and the high- or low-value distribution of coupling coordination degree in the GBA are not completely overlapped, they are related in spatial structure and time evolution. At present, the efficiency and scale of tourism are in a state of mutual restriction and mutual promotion at a medium level. It is necessary to adjust the allocation of resources according to local conditions and disperse the tourism resources, technology, talents, and other elements invested in the central city to other cities so as to improve the quality of the tourism economy and the effectiveness of regional integration [4].

## 5.2. Theoretical implications

Most of the existing studies on regional tourism development differences and spatial structure take one of the temporal evolution or structural characteristics of regional tourism differences and start with dynamic time series data or static spatial structure section data to analyze the spatial economic connection effect and collaborative development mechanism of regional tourism development, cutting the dynamic combination of the temporal and spatial attributes of regional tourism development. Therefore, it is limited to revealing the spatial-temporal differentiation and dynamic evolution of regional tourism. In order to address this issue, this study conducted some theoretical research on the spatial-temporal evolution and coupling analysis of tourism efficiency and tourism scale in the GBA.

It can enrich the research system in tourism economics. The first is to deepen the theory of tourism efficiency and scale. This study can further enrich and deepen the theoretical system of tourism efficiency and scale in tourism economics by analyzing the changes in the efficiency and scale of the tourism industry in the GBA. The second is to expand the application of spatiotemporal analysis methods. Using spatiotemporal analysis methods to study the dynamic evolution of tourism efficiency and scale provides a new perspective and tool for tourism economic research and helps to reveal the spatiotemporal patterns and characteristics of tourism economic development. The third is to promote interdisciplinary integration. This research combines theories and methods from multiple disciplines such as tourism, economics, and geography, promoting interdisciplinary integration and the comprehensiveness and systematicness of tourism economic research.

It can improve the theory of coordinated development of regional tourism. On the one hand, it reveals the collaborative mechanism of regional tourism. Through coupling and coordination research, the collaborative development mechanism of the tourism industry among cities in the GBA can be revealed, providing theoretical support for regional tourism integration. On the other hand, enriching the theory of regional tourism networks. This study helps deepen the understanding of the structure of regional tourism networks and provides a theoretical basis for constructing and optimizing regional tourism networks.

### 5.3.  Practical implications

The research findings have reference and illuminating implications for improving regional tourism economic efficiency and coordinated development.

Firstly, guide regional tourism planning and development. It can optimize tourism resource allocation. By analyzing the spatiotemporal changes in tourism efficiency and scale, the allocation of tourism resources in various cities in the GBA can be clarified, providing decision-making references for optimizing tourism resource allocation. It can promote the coordinated development of regional tourism. Based on the research results of coupling coordination, a more scientific and reasonable strategy for the coordinated development of regional tourism can be formulated to promote the overall development of the tourism industry in the GBA.

Secondly, improve the quality of tourism industry development. On the one hand, improve tourism efficiency. Studying the changing patterns of tourism efficiency can help identify bottlenecks in the development of the tourism industry, propose targeted improvement measures, and enhance the operational efficiency of the industry. On the other hand, promote the rational growth of the tourism scale. By analyzing the spatiotemporal changes in the tourism scale, it is possible to guide the rational growth of the tourism scale, avoid overdevelopment and disorderly competition, and ensure the sustainable development of the tourism industry.

Thirdly, promote regional economic integration. The first is to enhance regional competitiveness. The coordinated development of the tourism industry helps to enhance the overall competitiveness of the GBA, attract more domestic and foreign tourists, and promote regional economic development. The second is to promote the process of regional economic integration. As an important component of the regional economy, the coordinated development of the tourism industry can help promote the process of regional economic integration in the GBA and achieve coordinated development of the regional economy.

### 5.4.  Limitation and future research

The research area of this study was in southern China, so first, follow-up research can be conducted in different urban agglomerations to generalize the research. Secondly, in the future, we will further study the cities in the GBA that have improved their coordination in recent years, focusing on the influencing factors and causes that promote the improvement of their coordination, so as to provide a useful reference for the coordinated development of the tourism industry in other urban agglomerations.

## Acknowledgments

Thanks to all authors for their efforts in conducting this research.

Original map source is from https://nr.gd.gov.cn/map/bzdt/Index.aspx?area_click=8&area_level=#

## Author contributions

**Conceptualization:** Kaijun Wu.

**Data curation:** Kaijun Wu, Xilin Yang.

**Formal analysis:** Kaijun Wu.

**Funding acquisition:** Kaijun Wu.

**Methodology:** Kaijun Wu, Xilin Yang.

**Resources:** Kaijun Wu.

**Supervision:** Kaijun Wu.

**Validation:** Kaijun Wu.

**Visualization:** Xingfu Han.

**Writing – original draft:** Kaijun Wu, Xilin Yang.

**Writing – review & editing:** Kaijun Wu, Xingfu Han.

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
