## [Decision Letter · Decision Letter 0]

12 Apr 2024

PONE-D-24-04406Study on Temporal-spatial Changes and Tourism Efficiency Coupling in Guangdong-Hong Kong-Macao Greater Bay AreaPLOS ONE

Dear Dr. Wu,

Thank you for submitting your manuscript to PLOS ONE. After careful consideration, we feel that it has merit but does not fully meet PLOS ONE’s publication criteria as it currently stands. Therefore, we invite you to submit a revised version of the manuscript that addresses the points raised during the review process.

We look forward to receiving your revised manuscript.

Kind regards,

Xufeng Cui, Ph.D

Academic Editor

PLOS ONE

Journal Requirements:

"Guangdong Province Philosophy and Social Science Planning Project，Project number: GD23SQYJ03"

3. We note that your Data Availability Statement is currently as follows: All relevant data are within the manuscript and its Supporting Information files

Reviewers' comments:

Reviewer's Responses to Questions

**Comments to the Author**

1. Is the manuscript technically sound, and do the data support the conclusions?

Reviewer #1: Yes

Reviewer #2: Yes

2. Has the statistical analysis been performed appropriately and rigorously? 

Reviewer #1: Yes

Reviewer #2: No

3. Have the authors made all data underlying the findings in their manuscript fully available?

Reviewer #1: Yes

Reviewer #2: No

4. Is the manuscript presented in an intelligible fashion and written in standard English?

Reviewer #1: No

Reviewer #2: No

5. Review Comments to the Author

Reviewer #1: This study investigates the temporal-spatial changes and the coupling coordination relationship of tourism efficiency and scale in the GBA. It delves into the spatial-temporal disparities in tourism efficiency and scale from economic and geographic perspectives. The manuscript offers a model of coupled and coordinated development, and puts forth corresponding recommendations for regional tourism development in the GBA. The manuscript is well organized. However, some interpretations are missing. Given this, my recommendation is Major Revision. Bellows are my specific comments:

1. In Abstract, ESDA should be given a full name when it first appears.

2. Please check the entire manuscript. For nouns that require abbreviations, use the full name and indicate the abbreviation when the abstract and introduction first appear, and hold the abbreviation status in the following text.

3. In introduction, while the role and importance of the bay area economy and tourism in the global context are well-articulated. However, when describing the tourism resources and development level of the GBA, please provide specific examples or data.

4. The scientific literature can be enriched, particularly for socioeconomical development of the GBA. The following publication can be considered.

- He, Y., Wu, L., Liang, Y., Zheng, Y., Zhang, L. (2024). Spatial suitability between water supply pressure and waterworks water supply capacity in the Guangdong-Hong Kong-Macau Greater Bay Area: Spatial-temporal evolution pattern, driving mechanism and Implications. Journal of Cleaner Production, 434, 140317.

5. Please supplement the background information and tools of ESDA while describing in detail why these methods and results are selected.

6. In literature review, please add a brief summary of the content, methodology, and results of the individual studies in the literature, and provide your opinion on the similarities, differences, and strengths of this study in relation to your research.

7. Formulas and their numbers need to be unified alignment standards, such as formula center and number right alignment.

8. Figure 1 is odd, please consider restructure it.

9. Please check the expression of the proper noun in the text and write it uniform. e.g. “Lisa cluster diagram” or “LISA cluster map”?

10. Figure 2 is odd, please consider restructure it. You can put two small maps in a row for better visualization, add place names and compasses to the maps.

11. When describing the research methodology and tools, can you give more details on the design rationale and application methods?

12. In Figure 9, add labels and compasses to the maps.

13. Please provide brief background information and definitions of terms used in the paper the first time they appear. e.g. "Coupling Coordination Degree", "Comprehensive Efficiency", etc.

14. In conclusion and discussion, please engage in a more profound analysis of the results and compare them with existing literature.

15. Please thoroughly check and revise the expression and grammar of the entire manuscript. Your manuscript needs to be carefully edited by a professional English editor, paying special attention to English grammar, spelling and sentence structure.

Reviewer #2: The paper provides valuable insights into the temporal-spatial changes and tourism efficiency coupling in the Guangdong-Hong Kong-Macao Greater Bay Area. It addresses an important topic and offers a comprehensive analysis of tourism efficiency using the DEA-BCC model and the total factor productivity index model. The study's focus on collaborative development and marketing strategies for enhancing travel services quality is commendable. However, there are concerns regarding the lack of detailed methodology descriptions, limited evaluation studies, and the need for more comparisons with existing frameworks in the field. Addressing these points would further strengthen the paper's impact and credibility.

1. Can you provide more details on the specific data sources used for analyzing tourism efficiency and scale in the Guangdong-Hong Kong-Macao Greater Bay Area? How were these data collected and processed to ensure accuracy and reliability in the analysis?

2. Could you elaborate on the methodology employed to measure tourism efficiency using the DEA-BCC model and the total factor productivity index model? How were these models applied in the context of regional tourism development, and what were the key considerations in their implementation?

3. In the exploration of temporal and spatial differentiation of tourism efficiency and scale, what were the main findings regarding the evolution of these factors over the study period from 2001 to 2019? How did the coupling analysis reveal insights into the relationship between tourism efficiency and scale in the region?

4. Can you discuss the implications of the study's results on the collaborative development and marketing strategies for enhancing the quality of travel services in the Greater Bay Area? How do the findings contribute to the understanding of regional tourism economies and their impact on overall economic development?

5. Have you considered conducting sensitivity analyses or robustness checks to validate the stability and reliability of the results obtained from the DEA-BCC model and the total factor productivity index model? How do you address potential uncertainties or variations in the data that may affect the conclusions drawn from the study?

6. PLOS authors have the option to publish the peer review history of their article (what does this mean? ). If published, this will include your full peer review and any attached files.

**Do you want your identity to be public for this peer review?** For information about this choice, including consent withdrawal, please see our Privacy Policy .

Reviewer #1: No

Reviewer #2: No

---

## [Author Response · Author response to Decision Letter 0]

21 May 2024

Response to Reviewers’ comment

Manuscript No. PONE-D-24-04406

Title:Study on Temporal-spatial Changes and Tourism Efficiency Coupling in Guangdong-Hong Kong-Macao Greater Bay Area

Dear Editor,

We appreciate you very much for your positive and constructive comments on our manuscript. We have fully revised our manuscript and have addressed all of reviewer’s comments. The detailed revisions are listed below and highlighted in the revised manuscript with red background. We hope that the revisions in the resubmitted manuscript and our responses will be sufficient to make our manuscript meets your high standards.

Thank you for your consideration and time.

Sincerely yours,

Kaijun Wu*

International Cruise & Yacht College of Guangzhou Maritime University, Guangzhou 510725, China, gzwkj@126.com

Xilin Yang,

School of Psychology, Shenzhen University, Shenzhen 518000,China, 1711945825@qq.com

Xingfu Han

Culture Tourism & Geography School of Guangdong University of Finance & Economics, Guangzhou 510320, China, 2354780221@qq.com

Response to Reviewers’ comment

Manuscript No. PONE-D-24-04406

Title:Study on Temporal-spatial Changes and Tourism Efficiency Coupling in Guangdong-Hong Kong-Macao Greater Bay Area

Responses to reviewer 1:

First, we would like to thank you for your positive feedback of the paper and providing us very useful and constructive comments and suggestions. Listed below are the point-by-point responses to the reviews’ comments. We hope our revision may improve the paper. In the new version of our paper, the modified parts have been highlighted in red.

Comment 1: In Abstract, ESDA should be given a full name when it first appears.

Response: ESDA has be given a full name as “Earth Science Data Acquisition”.

Comment 2: Please check the entire manuscript. For nouns that require abbreviations, use the full name and indicate the abbreviation when the abstract and introduction first appear, and hold the abbreviation status in the following text.

Response: We have made modifications to the entire manuscript according to your suggestions.

Comment 3: In introduction, while the role and importance of the bay area economy and tourism in the global context are well-articulated. However, when describing the tourism resources and development level of the GBA, please provide specific examples or data.

Response: Increased data on inbound tourism and high-level tourism resources in GBA.

Comment 4: The scientific literature can be enriched, particularly for socioeconomical development of the GBA. The following publication can be considered.

Response: According to your suggestion, some literature related to GBA has been added, including the one you suggested.

Comment 5: Please supplement the background information and tools of ESDA while describing in detail why these methods and results are selected.

Response: 3.1.4 provides a detailed introduction to ESDA and an analysis of the reasons for choosing this method.

Comment 6: In literature review, please add a brief summary of the content, methodology, and results of the individual studies in the literature, and provide your opinion on the similarities, differences, and strengths of this study in relation to your research.

Response: The first two paragraphs of “3.Methodology” in Page6 provide a brief review of the literature and their reference and inspiration for this study.

Comment 7: Formulas and their numbers need to be unified alignment standards, such as formula center and number right alignment.

Response: We have made adjustments and standardized according to your suggestions.

Comment 8: Figure 1 is odd, please consider restructure it.

Response: We have made adjustments to Figure 1 according to your suggestion.

Comment 9: Please check the expression of the proper noun in the text and write it uniform. e.g. “Lisa cluster diagram” or “LISA cluster map”?

Response: We have rechecked the expression of specific nouns in the text according to your suggestion and made unified writing.

Comment 10: Figure 2 is odd, please consider restructure it. You can put two small maps in a row for better visualization, add place names and compasses to the maps.

Response: We have made adjustments according to your suggestion and added place names and compasses to the maps.

Comment 11: When describing the research methodology and tools, can you give more details on the design rationale and application methods?

Response:

(1)DEA-BCC model: Further detailed explanations were provided for the DEA-BCC model in 3.1.1.

(2)Malmquist productivity index model:Detailed introduction has been provided in 3.1.2.

(3)Exploratory spatial data analysis(ESDA):Detailed introduction has been provided in 3.1.3.

Comment 12: In Figure 9, add labels and compasses to the maps.

Response: We have made adjustments according to your suggestion and added place names and compasses to the maps.

Comment 13: Please provide brief background information and definitions of terms used in the paper the first time they appear. e.g. "Coupling Coordination Degree", "Comprehensive Efficiency", etc

Response: Part of the paper briefly introduces the background and definitions, such as:

Coupling and coordination model:

Coupling degree is an index to judge the degree of interaction between two or more elemental systems based on quantitative data (Wang Cheng, Tang Ning, 2018).(page9)

Coupling Coordination Degree is an index to quantitatively judge the goodness of coupling between two or more element systems, and it is a further analysis of the coupling state of tourism efficiency and scale.

Comment 14: In conclusion and discussion, please engage in a more profound analysis of the results and compare them with existing literature.

Response: Supplemented literature in similar Chinese contexts.

Comment 15: Please thoroughly check and revise the expression and grammar of the entire manuscript. Your manuscript needs to be carefully edited by a professional English editor, paying special attention to English grammar, spelling and sentence structure.

Response: A professional English editor has been invited to re edit the paper.

Responses to reviewer 2:

First, we would like to thank you for your positive feedback of the paper and providing us very useful and constructive comments and suggestions. Listed below are the point-by-point responses to the reviews’ comments. We hope our revision may improve the paper. In the new version of our paper, the modified parts have been highlighted in red.

Comment 1: Can you provide more details on the specific data sources used for analyzing tourism efficiency and scale in the Guangdong-Hong Kong-Macao Greater Bay Area? How were these data collected and processed to ensure accuracy and reliability in the analysis?

Response:The specific data sources are detailed in 3.2.2, and the selection of indicators is detailed in 3.2.1.

Comment 2: Could you elaborate on the methodology employed to measure tourism efficiency using the DEA-BCC model and the total factor productivity index model? How were these models applied in the context of regional tourism development, and what were the key considerations in their implementation?

Response: Further detailed explanations were provided for the DEA-BCC model in 3.1.1.

The total factor productivity index model is calculated using the DEA-Malmquist productivity index model. The detailed calculation process of the DEA Malmquist productivity index model is described in 3.1.2.

Comment 3: In the exploration of temporal and spatial differentiation of tourism efficiency and scale, what were the main findings regarding the evolution of these factors over the study period from 2001 to 2019? How did the coupling analysis reveal insights into the relationship between tourism efficiency and scale in the region?

Response:The main findings of this study were analyzed in detail in section 4. Results from 11 to 25. The spatiotemporal differences in tourism efficiency and scale in GBA are mainly analyzed from the static characteristics of tourism efficiency and scale, the dynamic characteristics of tourism efficiency, the dynamic characteristics of tourism scale, the spatiotemporal dynamic changes of tourism efficiency, and the spatiotemporal dynamic changes of tourism scale. The coupling and coordination relationship between tourism efficiency and tourism scale in GBA is mainly analyzed from the aspects of coupling degree spatiotemporal differentiation and coupling coordination degree spatiotemporal differentiation.

A brief summary of the results is presented in 5.1 Conclusions.

Comment 4: Can you discuss the implications of the study's results on the collaborative development and marketing strategies for enhancing the quality of travel services in the Greater Bay Area? How do the findings contribute to the understanding of regional tourism economies and their impact on overall economic development?

Response:Your proposal is of great help to the subsequent research on tourism marketing in the GBA, but the theme of this article is to explore the spatiotemporal changes and coordinated development mechanisms of 11 cities in GBA through historical data.

As for how to help understand the regional tourism economies and their impact on overall economic development, it is stated in the Practical implications.

Comment 5: Have you considered conducting sensitivity analyses or robustness checks to validate the stability and reliability of the results obtained from the DEA-BCC model and the total factor productivity index model? How do you address potential uncertainties or variations in the data that may affect the conclusions drawn from the study?

Response:The robustness test of panel data is supplemented in section 3.2.3.

---

## [Decision Letter · Decision Letter 1]

7 Jul 2024

PONE-D-24-04406R1Study on Temporal-spatial Changes and Tourism Efficiency Coupling in Guangdong-Hong Kong-Macao Greater Bay AreaPLOS ONE

Dear Dr. Wu,

Thank you for submitting your manuscript to PLOS ONE. After careful consideration, we feel that it has merit but does not fully meet PLOS ONE’s publication criteria as it currently stands. Therefore, we invite you to submit a revised version of the manuscript that addresses the points raised during the review process.

We look forward to receiving your revised manuscript.

Kind regards,

Xufeng Cui, Ph.D

Academic Editor

PLOS ONE

Reviewers' comments:

Reviewer's Responses to Questions

**Comments to the Author**

1. If the authors have adequately addressed your comments raised in a previous round of review and you feel that this manuscript is now acceptable for publication, you may indicate that here to bypass the “Comments to the Author” section, enter your conflict of interest statement in the “Confidential to Editor” section, and submit your "Accept" recommendation.

Reviewer #1: (No Response)

Reviewer #3: (No Response)

2. Is the manuscript technically sound, and do the data support the conclusions?

Reviewer #1: Partly

Reviewer #3: Yes

3. Has the statistical analysis been performed appropriately and rigorously? 

Reviewer #1: Yes

Reviewer #3: Yes

4. Have the authors made all data underlying the findings in their manuscript fully available?

Reviewer #1: No

Reviewer #3: Yes

5. Is the manuscript presented in an intelligible fashion and written in standard English?

Reviewer #1: No

Reviewer #3: Yes

6. Review Comments to the Author

Reviewer #1: The authors made some improvements, however, references of the revised version should be appropriately checked and improved.

Reviewer #3: This paper has certain theoretical and practical significance, and its structure is relatively complete, but there are still some problems that need to be improved:

1. Some of the words used are inconsistent, such as the introduction part “the coupled and coordinated development model” is inconsistent with similar expressions in the abstract.

2. The research methodology is relatively common, so it is not necessary to introduce it at length.

3. The results of the empirical analyses lack a brief analysis of the causes.

4. The authors should double-check the coupling formula.

5. On what basis is the grading of coupling coordination degree classified? Please give the basis.

6. The images layout must be further adjusted, for example, the size and position of Fig. 9(b) fails to be consistent with several other figures, please make reasonable adjustments.

7. Indicators are cited with reasons, but the reference articles are not clearly indicated.

8. The expression of the conclusion is not concise enough, and some of the contents of the conclusion can be incorporated into the discussion.

9. “Theoretical impact” in the discussion is to express the similarities and differences between this article and related studies?

10. The innovative points or marginal contributions of the article need to be more clearly reflected.

7. PLOS authors have the option to publish the peer review history of their article (what does this mean? ). If published, this will include your full peer review and any attached files.

**Do you want your identity to be public for this peer review?** For information about this choice, including consent withdrawal, please see our Privacy Policy .

Reviewer #1: No

Reviewer #3: No

---

## [Author Response · Author response to Decision Letter 1]

20 Jul 2024

Dear Editor,

We appreciate you very much for your positive and constructive comments on our manuscript. We have fully revised our manuscript and have addressed all of reviewer’s comments. The detailed revisions are listed below and highlighted in the revised manuscript with red background. We hope that the revisions in the resubmitted manuscript and our responses will be sufficient to make our manuscript meets your high standards.

Thank you for your consideration and time.

Sincerely yours,

Kaijun Wu*

International Cruise & Yacht College of Guangzhou Maritime University, Guangzhou 510725, China, gzwkj@126.com

Xilin Yang,

School of Psychology, Shenzhen University, Shenzhen 518000,China, 1711945825@qq.com

Xingfu Han

Culture Tourism & Geography School of Guangdong University of Finance & Economics, Guangzhou 510320, China, 2354780221@qq.com

Response to Editor’ comment

Manuscript No. PONE-D-24-04406

Title:Study on Temporal-spatial Changes and Tourism Efficiency Coupling in Guangdong-Hong Kong-Macao Greater Bay Area

Responses to editor:

First, we would like to thank you for your positive feedback of the paper and providing us very useful and constructive comments and suggestions again. Listed below are the point-by-point responses to editor’ comments. We hope our second revision may improve the paper. In the new version of our paper, the modified parts have been highlighted in blue.

Comment 1: About the data available on acceptance.

Response: We have provided a data sharing link.

link: https://pan.baidu.com/s/14SaDRfgjn3mTyZSWsMgooQ Extracted code: 2azu

Comment 2: About figure 2, 6, 8 and 9 in your submission.

Response: Because we were unable to obtain permission, we changed the corresponding figures to be presented in the form of tables. Correspondingly, convert Figure 2 into Table 2, Figure 6 into Table 4, Figure 8 into Table 5, and Figure 9 into Table 6.

Response to Reviewers’ comment

Manuscript No. PONE-D-24-04406

Title:Study on Temporal-spatial Changes and Tourism Efficiency Coupling in Guangdong-Hong Kong-Macao Greater Bay Area

Responses to reviewer 1:

First, we would like to thank you for your positive feedback of the paper and providing us very useful and constructive comments and suggestions again. Listed below are the point-by-point responses to the reviews’ comments. We hope our second revision may improve the paper. In the new version of our paper, the modified parts have been highlighted in red.

Comment 1: The authors made some improvements, however, references of the revised version should be appropriately checked and improved.

Response: We have carefully checked and revised the references.

Responses to reviewer 3:

First, we would like to thank you for your positive feedback of the paper and providing us very useful and constructive comments and suggestions again. Listed below are the point-by-point responses to the reviews’ comments. We hope our second revision may improve the paper. In the new version of our paper, the modified parts have been highlighted in green.

Comment 1: Some of the words used are inconsistent, such as the introduction part “the coupled and coordinated development model” is inconsistent with similar expressions in the abstract.

Response: We have carefully examined the inconsistent expressions in the text and have unified them.

Comment 2: The research methodology is relatively common, so it is not necessary to introduce it at length.

Response: During the revision of the first draft, Reviewer 1 suggested that some research methods require more detailed explanations, so additional modifications were made based on the original manuscript.

Comment 3: The results of the empirical analyses lack a brief analysis of the causes.

Response: The results of the empirical analysis actually include a brief analysis of the causes, such as the possible reasons for the results of some cities in 4.1.1. 1) in green font.

Comment 4: The authors should double-check the coupling formula.

Response: We have carefully checked the coupling degree formula and coupling coordination degree model again, and there should be no problem. If there are any further errors, please point them out and we will make the necessary revisions.

Comment 5: On what basis is the grading of coupling coordination degree classified? Please give the basis.

Response:We have added the grading criteria for coupling coordination degree in the last paragraph of 3.1.5.

Comment 6: The images layout must be further adjusted, for example, the size and position of Fig. 9(b) fails to be consistent with several other figures, please make reasonable adjustments.

Response: We make reasonable adjustments to the layout such as image size and position.

Comment 7: Indicators are cited with reasons, but the reference articles are not clearly indicated.

Response: We have added reference annotations for the indicators.

Comment 8: The expression of the conclusion is not concise enough, and some of the contents of the conclusion can be incorporated into the discussion.

Response:Based on your suggestion, we have simplified the expression of the conclusion.

Comment 9: “Theoretical impact” in the discussion is to express the similarities and differences between this article and related studies?

Response:There is no 'Theoretical impact' in the manuscript, only 'Theoretical implications'. We further refined the theoretical and practical implications.

Comment 10: The innovative points or marginal contributions of the article need to be more clearly reflected.

Response:We further refined the theoretical and practical implications.

---

## [Decision Letter · Decision Letter 2]

16 Aug 2024

PONE-D-24-04406R2Study on Temporal-spatial Changes and Tourism Efficiency Coupling in Guangdong-Hong Kong-Macao Greater Bay AreaPLOS ONE

Dear Dr. Wu,

Thank you for submitting your manuscript to PLOS ONE. After careful consideration, we feel that it has merit but does not fully meet PLOS ONE’s publication criteria as it currently stands. Therefore, we invite you to submit a revised version of the manuscript that addresses the points raised during the review process. Drop "study on" from the title.

**ACADEMIC EDITOR:** The author excessively self-cites, and irrelevant self-citations need to be removed. A thorough revision is required; otherwise, the paper cannot be accepted. The map creation in the paper is not up to standard; it is recommended that the author seek review from a scholar with a background in cartography.The academic rigor of the paper needs to be improved.

We look forward to receiving your revised manuscript.

Kind regards,

Xufeng Cui, Ph.D

Academic Editor

PLOS ONE

Additional Editor Comments:

The author excessively self-cites, and irrelevant self-citations need to be removed. A thorough revision is required; otherwise, the paper cannot be accepted. The map creation in the paper is not up to standard; it is recommended that the author seek review from a scholar with a background in cartography.The academic rigor of the paper needs to be improved.

Reviewers' comments:

Reviewer's Responses to Questions

**Comments to the Author**

1. If the authors have adequately addressed your comments raised in a previous round of review and you feel that this manuscript is now acceptable for publication, you may indicate that here to bypass the “Comments to the Author” section, enter your conflict of interest statement in the “Confidential to Editor” section, and submit your "Accept" recommendation.

Reviewer #1: (No Response)

Reviewer #3: All comments have been addressed

2. Is the manuscript technically sound, and do the data support the conclusions?

Reviewer #1: Partly

Reviewer #3: Yes

3. Has the statistical analysis been performed appropriately and rigorously? 

Reviewer #1: Yes

Reviewer #3: Yes

4. Have the authors made all data underlying the findings in their manuscript fully available?

Reviewer #1: Yes

Reviewer #3: Yes

5. Is the manuscript presented in an intelligible fashion and written in standard English?

Reviewer #1: No

Reviewer #3: Yes

6. Review Comments to the Author

Reviewer #1: (No Response)

Reviewer #3: The authors have substantially revised and optimized this manuscript to meet the requirements for publication in this journal.

7. PLOS authors have the option to publish the peer review history of their article (what does this mean? ). If published, this will include your full peer review and any attached files.

**Do you want your identity to be public for this peer review?** For information about this choice, including consent withdrawal, please see our Privacy Policy .

Reviewer #1: No

Reviewer #3: No

---

## [Author Response · Author response to Decision Letter 2]

20 Aug 2024

Response to Reviewers’ comment

Manuscript No. PONE-D-24-04406

Title:Study on Temporal-spatial Changes and Tourism Efficiency Coupling in Guangdong-Hong Kong-Macao Greater Bay Area

Dear Editor,

We appreciate you very much for your positive and constructive comments on our manuscript. We have fully revised our manuscript and have addressed all of reviewer’s comments. The detailed revisions are listed below and highlighted in the revised manuscript with red background. We hope that the revisions in the resubmitted manuscript and our responses will be sufficient to make our manuscript meets your high standards.

Thank you for your consideration and time.

Sincerely yours,

Kaijun Wu*

International Cruise & Yacht College, Guangzhou Maritime University, Guangzhou 510725, China, gzwkj@126.com

Binlan Wu*

School of Economics and Management, South China Normal University, Guangzhou 510000, China, binglanwu@foxmail.com

Xilin Yang,

School of Psychology, Shenzhen University, Shenzhen 518000, China, 1711945825@qq.com

Wanfu Jin

School of Geography and Environmental Economics, Guangdong University of Finance & Economics, Guangzhou 510320, China, jinwanfu0927@163.com

Response to Reviewers’ comment

Manuscript No. PONE-D-24-04406

Title:Study on Temporal-spatial Changes and Tourism Efficiency Coupling in Guangdong-Hong Kong-Macao Greater Bay Area

Responses to editor:

First, we would like to thank you for your positive feedback of the paper and providing us very useful and constructive comments and suggestions again. Listed below are the point-by-point responses to the editor’ comments. We hope our fourth revision may improve the paper.

Comment 1: Drop "study on" from the title.

Response: We have dropped them.

Comment 2: The author excessively self-cites, and irrelevant self-citations need to be removed.

Response: We have removed references 47, 48, 49.

Comment 3: A thorough revision is required.

Response: We have made a one-step revision and improvement to the paper.

Comment 4: The map creation in the paper is not up to standard; it is recommended that the author seek review from a scholar with a background in cartography.

Response: We asked scholars with a background in geography to remake the map and clearly indicate the source of the original map in Acknowledgments.

Comment 5: The academic rigor of the paper needs to be improved.

Response: We have further improved the academic rigor of the paper.

Responses to reviewer 1:

First, we would like to thank you for your positive feedback of the paper and providing us very useful and constructive comments and suggestions again. Listed below are the point-by-point responses to the reviews’ comments. We hope our fourth revision may improve the paper. In the new version of our paper, the modified parts have been highlighted in red.

Comment 1: The authors made some improvements, however, references of the revised version should be appropriately checked and improved.

Response: We have carefully checked and revised the references.

Responses to reviewer 3:

First, we would like to thank you for your positive feedback of the paper and providing us very useful and constructive comments and suggestions again. Listed below are the point-by-point responses to the reviews’ comments. We hope our fourth revision may improve the paper. In the new version of our paper, the modified parts have been highlighted in green.

Comment 1: Some of the words used are inconsistent, such as the introduction part “the coupled and coordinated development model” is inconsistent with similar expressions in the abstract.

Response: We have carefully examined the inconsistent expressions in the text and have unified them.

Comment 2: The research methodology is relatively common, so it is not necessary to introduce it at length.

Response: During the revision of the first draft, Reviewer 1 suggested that some research methods require more detailed explanations, so additional modifications were made based on the original manuscript.

Comment 3: The results of the empirical analyses lack a brief analysis of the causes.

Response: The results of the empirical analysis actually include a brief analysis of the causes, such as the possible reasons for the results of some cities in 4.1.1. 1) in green font.

Comment 4: The authors should double-check the coupling formula.

Response: We have carefully checked the coupling degree formula and coupling coordination degree model again, and there should be no problem. If there are any further errors, please point them out and we will make the necessary revisions.

Comment 5: On what basis is the grading of coupling coordination degree classified? Please give the basis.

Response:We have added the grading criteria for coupling coordination degree in the last paragraph of 3.1.5.

Comment 6: The images layout must be further adjusted, for example, the size and position of Fig. 9(b) fails to be consistent with several other figures, please make reasonable adjustments.

Response: We make reasonable adjustments to the layout such as image size and position.

Comment 7: Indicators are cited with reasons, but the reference articles are not clearly indicated.

Response: We have added reference annotations for the indicators.

Comment 8: The expression of the conclusion is not concise enough, and some of the contents of the conclusion can be incorporated into the discussion.

Response:Based on your suggestion, we have simplified the expression of the conclusion.

Comment 9: “Theoretical impact” in the discussion is to express the similarities and differences between this article and related studies?

Response:There is no 'Theoretical impact' in the manuscript, only 'Theoretical implications'. We further refined the theoretical and practical implications.

Comment 10: The innovative points or marginal contributions of the article need to be more clearly reflected.

Response:We further refined the theoretical and practical implications.

---

## [Decision Letter · Decision Letter 3]

9 Sep 2024

PONE-D-24-04406R3Temporal-spatial Changes and Tourism Efficiency Coupling in Guangdong-Hong Kong-Macao Greater Bay AreaPLOS ONE

Dear Dr. Wu,

Thank you for submitting your manuscript to PLOS ONE. After careful consideration, we feel that it has merit but does not fully meet PLOS ONE’s publication criteria as it currently stands. Therefore, we invite you to submit a revised version of the manuscript that addresses the points raised during the review process.

The editorial team has noticed that one author has been removed and two new authors have been added to the revised manuscript.In accordance with the journal's academic standards, we require a detailed account of the contributions made by this new author to the manuscript.

Please provide a thorough explanation of their specific contributions to the manuscript revision, along with any supporting documentation or evidence that demonstrates their involvement in the research and writing process. It is important to ensure that all contributors' roles are clearly articulated and documented.

Please be aware that academic misconduct, including the improper addition of authors, may result in the manuscript being rejected. We urge you to review the journal's guidelines on authorship and academic integrity to ensure compliance.

We look forward to receiving your revised manuscript.

Kind regards,

Xufeng Cui, Ph.D

Academic Editor

PLOS ONE

Additional Editor Comments:

The editorial team has noticed that one author has been removed and two new authors have been added to the revised manuscript.In accordance with the journal's academic standards, we require a detailed account of the contributions made by this new author to the manuscript.

Please provide a thorough explanation of their specific contributions to the manuscript revision, along with any supporting documentation or evidence that demonstrates their involvement in the research and writing process. It is important to ensure that all contributors' roles are clearly articulated and documented.

Please be aware that academic misconduct, including the improper addition of authors, may result in the manuscript being rejected. We urge you to review the journal's guidelines on authorship and academic integrity to ensure compliance.

---

## [Author Response · Author response to Decision Letter 3]

22 Oct 2024

Response to Reviewers’ comment

Manuscript No. PONE-D-24-04406

Title:Temporal-spatial Changes and Tourism Efficiency Coupling in Guangdong-Hong Kong-Macao Greater Bay Area

Dear Editor,

We appreciate you very much for your positive and constructive comments on our manuscript. We have fully revised our manuscript and have addressed all of your comments. The detailed revisions are listed below and highlighted in the revised manuscript with a red background. We hope that the revisions in the resubmitted manuscript and our responses will be sufficient to make our manuscript meet your high standards.

Thank you for your consideration and time.

Sincerely yours,

Kaijun Wu*

School of Maritime Law and Traffic Management, Guangzhou Maritime University, Guangzhou 510725, China, gzwkj@126.com

Xilin Yang,

School of Psychology, Shenzhen University, Shenzhen 518000, China, 1711945825@qq.com

Xingfu Han

Huawei Institute of Information, Communication and Technology，Guangdong Baiyun University, Guangzhou 510450, China, 2354780221@qq.com

Response to Reviewers’ comment

Manuscript No. PONE-D-24-04406

Title: Temporal-spatial Changes and Tourism Efficiency Coupling in Guangdong-Hong Kong-Macao Greater Bay Area

Responses to editor:

First, we would like to thank you for your positive feedback on the paper and for providing us very useful and constructive comments and suggestions again. Listed below are the point-by-point responses to the editor’s comments. We hope our tenth revision may improve the paper.

Comment 1: One of the provided URLs, http://stats.gd.gov.cn/gdtjnj/index.html, remains non-functional for the journal office. Accessing this URL yields only a "500 Internal Server Error.".

Response:

Solution 1: I have confirmed once again that this website is valid, and the page it opens is shown in the screenshot. Alternatively, you can try opening it with another browser.

The data collection year is from 2001 to 2019. I randomly click on one of the year links, such as 2001, which can also enter the webpage, as shown in the screenshot below.

Solution 2: Do not use this network source, as in 3.2.2 The data sources have clearly stated that the relevant data comes from the "Statistical Yearbook of Guangdong Province". There are web versions of the Statistical Yearbook of Guangdong Province, as well as printed books, and their data is the same.

Solution 3: Since the data collection years are from 2001 to 2019, I am providing you with all the links for these 19 years to see if they can be opened.

2001: http://stats.gd.gov.cn/gdtjnj/content/post_1424898.html

2002: http://stats.gd.gov.cn/gdtjnj/content/post_1424899.html

2003: http://stats.gd.gov.cn/gdtjnj/content/post_1424900.html

2004: http://stats.gd.gov.cn/gdtjnj/content/post_1424901.html

2005: http://stats.gd.gov.cn/gdtjnj/content/post_1424902.html

2006: http://stats.gd.gov.cn/gdtjnj/content/post_1424885.html

2007: http://stats.gd.gov.cn/gdtjnj/content/post_1424892.html

2008: http://stats.gd.gov.cn/gdtjnj/content/post_1424886.html

2009: http://stats.gd.gov.cn/gdtjnj/content/post_1424887.html

2010: http://stats.gd.gov.cn/gdtjnj/content/post_1424888.html

2011: http://stats.gd.gov.cn/gdtjnj/content/post_1424889.html

2012: http://stats.gd.gov.cn/gdtjnj/content/post_1424890.html

2013: http://stats.gd.gov.cn/gdtjnj/content/post_1424891.html

2014: http://stats.gd.gov.cn/gdtjnj/content/post_1424893.html

2015: http://stats.gd.gov.cn/gdtjnj/content/post_1424894.html

2016: http://stats.gd.gov.cn/gdtjnj/content/post_1424895.html

2017: http://stats.gd.gov.cn/gdtjnj/content/post_1424896.html

2018: http://stats.gd.gov.cn/gdtjnj/content/post_1424903.html

2019: http://stats.gd.gov.cn/gdtjnj/content/post_2639622.html

---

## [Decision Letter · Decision Letter 4]

5 Nov 2024

Temporal-spatial Changes and Tourism Efficiency Coupling in Guangdong-Hong Kong-Macao Greater Bay Area

PONE-D-24-04406R4

Dear Dr. Wu,

We’re pleased to inform you that your manuscript has been judged scientifically suitable for publication and will be formally accepted for publication once it meets all outstanding technical requirements.

Kind regards,

Xufeng Cui, Ph.D

Academic Editor

PLOS ONE

Additional Editor Comments (optional):

It is recommended for acceptance. 

It is recommended to improve the paper's formatting in accordance with the journal's formatting requirements.
---

## [Editor Report · Acceptance letter]

PONE-D-24-04406R4

PLOS ONE

Dear Dr. Wu,

I'm pleased to inform you that your manuscript has been deemed suitable for publication in PLOS ONE. Congratulations! Your manuscript is now being handed over to our production team.

Kind regards,

on behalf of

Professor Xufeng Cui

Academic Editor

PLOS ONE